**LABORATORY EXPERIMENTAL INVESTIGATION OF HEAT TRANSPORT IN**
**FRACTURED MEDIA**
Claudia Cherubini (1) (2), Nicola Pastore (3), Concetta I. Giasi (3), Nicoletta Maria Allegretti (3)
(1) Department of Mechanical, Aerospace & Civil Engineering - Brunel University London,
Uxbridge, UB8 3PH, United Kingdom (2) School of Civil Engineering, The University of
Queensland, Queensland, Australia (3) DICATECh - Department of Civil, Environmental, Building
Engineering, and Chemistry – Politecnico di Bari, Italy
The authors declares that there is no conflict of interest regarding the publication of this paper.

## 9 Abstract

Low enthalpy geothermal energy is a renewable resource that is still underexploited nowadays, in
relation to its potential for development in the society worldwide. Most of its applications have
already been investigated, such as: heating and cooling of private and public buildings, roads
defrost, cooling of industrial processes, food drying systems or desalination.
Geothermal power development is a long, risky and expensive process. It basically consists of
successive development stages aimed at locating the resources (exploration), confirming the power
generating capacity of the reservoir (confirmation) and building the power plant and associated
structures (site development). Different factors intervene in influencing the length, difficulty and
materials required for these phases thereby affecting their cost.
One of the major limitations related to the installation of low enthalpy geothermal power plants
regards the initial development steps which are risky and the upfront capital costs that are huge.
Most of the total cost of geothermal power is related to the reimbursement of invested capital and
associated returns.
In order to increase the optimal efficiency of installations which use groundwater as geothermal
resource, flow and heat transport dynamics in aquifers need to be well characterized. Especially in
fractured rock aquifers these processes represent critical elements that are not well known.
Therefore there is a tendency to oversize geothermal plants.
In literature there are very few studies on heat transport especially in fractured media.
This study is aimed to deepen the understanding of this topic through heat transport experiments in
fractured network and their interpretation.
Heat transfer tests have been carried out on the experimental apparatus previously employed to
perform flow and tracer transport experiments, which has been modified in order to analyze heat
transport dynamics in a network of fractures. In order to model the obtained thermal breakthrough

curves, the Explicit Network Model (ENM) has been used, which is based on an adaptation of a Tang's solution for the transport of the solutes in a semi-infinite single fracture embedded in a porous matrix.

Parameter estimation, time moment analysis, tailing character and other dimensionless parameters have permitted to better understand the dynamics of heat transport and the efficiency of heat exchange between the fractures and matrix. The results have been compared with the previous experimental studies on solute transport.

## 1 Introduction

An important role in transport of natural resources or contaminant transport through subsurface systems is given by fractured rocks. The interest about the study of dynamics of heat transport in fractured media has grown in recent years because of the development of a wide range of applications, including geothermal energy harvesting (Gisladottir et al., 2016).

Quantitative geothermal reservoir characterization using tracers is based on different approaches for predicting thermal breakthrough curves in fractured reservoirs (Shook, 2001, Kocabas, 2005, Read et al., 2013).

The characterization and modeling of heat transfer in fractured media is particularly challenging as open and well-connected fractures can induce highly localized pathways which are orders of magnitude more permeable than the rock matrix (Klepikova et al., 2016, Cherubini and Pastore, 2011).

The study of solute transport in fractured media has become recently a widely diffused research topic in hydrogeology (Cherubini, 2008, Cherubini et al., 2008, Cherubini et al., 2009, Cherubini et al., 2013d, Masciopinto et al., 2010), whereas the literature about heat transfer in fractured media is somewhat limited.

Hao et al. (2013) developed a dual continuum model for the representation of discrete fractures and the interaction with surrounding rock matrix in order to give a reliable prediction of the impacts of fracture – matrix interaction on heat transfer in fractured geothermal formations.

Moonen et al. (2011) introduced the concept of cohesive zone which represents a transition zone between the fracture and undamaged material. They proposed a model to adequately represent the influences of fractures or partially damaged material interfaces on heat transfer phenomena.

Geiger and Emmanuel (2010) found that matrix permeability plays an important role on thermal retardations and attenuation of thermal signal. At high matrix permeability, poorly connected fractures can contribute to the heat transport, resulting in heterogeneous heat distributions in the

whole matrix block. For lower matrix permeability heat transport occurs mainly through fractures that form a fully connected pathway between the inflow and outflow boundaries, that results in highly non – Fourier behavior, characterized by early breakthrough and long tailing.

Numerous field observations (Tsang and Neretnieks, 1998) show that flow in fractures is being organized in channels due to the small scale variations in the fracture aperture. Flow channeling causes dispersion in fractures. Such channels will have a strong influence on the transport characteristics of a fracture, such as, for instance, its thermal exchange area, crucial for geothermal applications (Auradou et al., 2006). Highly channelized flow in fractured geologic systems has been credited with early thermal breakthrough and poor performance of geothermal circulation systems (Hawkins et al., 2012).

Lu et. al (2012) conducted experiments of saturated water flow and heat transfer in a regularly fractured granite at meter scale. The experiments indicated that the heat advection due to water flow in vertical fractures nearest to the heat sources played a major role in influencing the spatial distributions and temporal variations of the temperature, impeding heat conduction in transverse direction; such effect increased with larger water fluxes in the fractures and decreased with higher heat source and/or larger distance of the fracture from the heat source.

Neuville et al. (2010) showed that fracture – matrix thermal exchange is highly affected by the fracture wall roughness. Natarajan et. al (2010) conducted numerical simulation of thermal transport in a sinusoidal fracture matrix coupled system. They affirmed that this model presents a different behavior respect to the classical parallel plate fracture matrix coupled system. The sinusoidal curvature of the fracture provides high thermal diffusion into the rock matrix.

Ouyang (2014) developed a three – equation local thermal non – equilibrium model to predict the effective solid – to – fluid heat transfer coefficient in geothermal system reservoirs. They affirmed that due to the high rock – to – fracture size ratio, the solid thermal resistance effect in the internal rocks cannot be neglected in the effective solid – to fluid heat transfer coefficient. Furthermore the results of this study show that it is not efficient to extract the thermal energy from the rocks if fracture density is not large enough.

Analytical and semi-analytical approaches have been developed to describe the dynamics of heat transfer in fractured rocks. Such approaches are amenable to the same mathematical treatment as their counterparts developed for mass transport (Martinez et al., 2014). One of these is the analytical solution derived by Tang et al. (1981).

While the equations of solute and thermal transport have the same basic form, the fundamental
difference between mass and heat transport is that: 1) solutes are transported through the fractures
only, whereas heat is transported through both fractures and matrix, 2) the fracture-matrix exchange
is large compared with molecular diffusion. This means that the fracture matrix exchange is more
relevant for heat transport than for mass transport. Thus, matrix thermal diffusivity strongly
influences the thermal breakthrough curves (BTCs) (Becker and Shapiro, 2003).
Contrarily, since the heat capacity of the solids will retard the advance of the thermal front, the
advective transport for heat is slower than for solute transport (Rau et al., 2012).
The quantification of thermal dispersivity as far as heat transport and its relationship with velocity
hasn't been properly addressed experimentally and has got conflicting descriptions in literature (Ma
et al., 2012).
Most studies neglect the hydrodynamic component of thermal dispersion because of thermal
diffusion being more efficient than molecular diffusion by several orders of magnitude (Bear 1972).
Analysis of heat transport under natural gradients has commonly neglected hydrodynamic
dispersion (e.g., Bredehoeft and Papadopulos, 1965; Domenico and Palciauskas, 1973; Taniguchi et
al., 1999; Reiter, 2001; Ferguson et al., 2006). Dispersive heat transport is often assumed to be
represented by thermal conductivity and/or to have little influence in models of relatively large
systems and modest fluid flow rates (Bear, 1972, Woodbury and Smith, 1985).
Some authors suggest that thermal dispersivity enhances the spreading of thermal energy and
should therefore be part of the mathematical description of heat transfer in analogy to solute
dispersivity (de Marsily, 1986) and have incorporated this term into their models (e.g., Smith and
Chapman, 1983; Hopmans et al., 2002; Niswonger and Prudic, 2003). In the same way, other
researchers (e.g., Smith and Chapman, 1983, Ronan et al., 1998, Constanz et al., 2002, Su et al.,
2004) have included the thermomechanical dispersion tensor representing mechanical mixing
caused by unspecified heterogeneities within the porous medium.
On the contrary, some other researchers argue that the enhanced thermal spreading is either
negligible or can be described simply by increasing the effective diffusivity, thus the hydrodynamic
dispersivity mechanism is inappropriate (Bear, 1972; Bravo et al., 2002, Ingebritsen and Sanford,
1998, Keery et al, 2007). Constantz et al. (2003) and Vandenbohede et al. (2009) found that thermal
dispersivity was significantly smaller than the solute dispersivity. Others (de Marsily, 1986,
Molina-Giraldo et al., 2011) found that thermal and solute dispersivity were on the same order of
magnitude.
Tracer tests of both solute and heat were carried out at Bonnaud, Jura, France (deMarsily, 1986) and
the thermal dispersivity and solute dispersivity were found of the same order of magnitude.
Bear (1972), Ingebritsen and Sanford (1998), and Hopmans et al. (2002), among others, concluded
that the effects of thermal dispersion are negligible compared to conduction and set the former to
zero.
However, Hopmans et al (2002) showed that dispersivity is increasingly important at higher flow
water velocities, since it is only then that the thermal dispersion term is of the same order of
magnitude or larger than the conductive term.
Sauty et al. (1982) suggested that there was a correlation between the apparent thermal conductivity
and Darcy velocity thus they included the hydrodynamic dispersion term in the advective-
conductive modeling.
Other similar formulations of this concept are present in the literature (e.g., Papadopulos and
Larson, 1978; Smith and Chapman, 1983; Molson et al., 1992). Such treatments have not explicitly
distinguished between macrodispersion, which occurs due to variations in permeability over larger
scales and the components of hydrodynamic dispersion that occur due to variations in velocity at
the pore scale.
One group of authors have utilized a linear relationship to describe the thermal dispersivity and the
relationship between thermal dispersivity and fluid velocity (e.g., de Marsily, 1986; Anderson,
2005; Hatch et al., 2006; Keery et al., 2007; Vandenbohede et al., 2009; Vandenbohede and Lebbe,
2010; Rau et al., 2010), while others have identified the possibility of a nonlinear relationship
(Green et al., 1964).
The present study is aimed at providing a better understanding of heat transfer mechanisms in
fractured rocks. Laboratory experiments on mass and heat transport in a fractured rock sample have
been carried out in order to analyze the contribution of thermal dispersion in heat propagation
processes, the influence of nonlinear flow dynamics on the enhancement of thermal matrix diffusion
and finally the optimal conditions for thermal exchange in a fractured network.
Section 1 shows a short review about mass and heat transport in fractured media highlighting what
is still unresolved or contrasting in the literature.
In Section 2 the theoretical background related to non linear flow, solute and heat transport behavior
in fractured media has been reported.
A better development of the Explicit Network Model (ENM), based on a Tang's solution developed
for solute transport in a single semi-infinite fracture inside a porous matrix has been used for the
fitting of the thermal BTCs. The ENM model explicitly takes the fracture network geometry into
account and therefore permits to understand the physical meaning of mass and heat transfer
phenomena and to obtain a more accurate estimation of the related parameters. In analogous way
the ENM model has been used in order to fit the observed BTCs obtained from previous
experiments on mass transport.
Section 3 shows the thermal tracer tests carried out on an artificially created fractured rock sample
that has been used in previous studies to analyze nonlinear flow and non Fickian transport dynamics
in fractured formations (Cherubini et al., 2012, 2013a, 2013b, 2013c and 2014).
In Section 4 have been reported the interpretation of flow and transport experiments together with
the fitting of BTCs and interpretation of estimated model parameters. In particular, the obtained
thermal BTCs show a more enhanced early arrival and long tailing than solute BTCs.
The travel time for solute transport is an order of magnitude lower than for heat transport
experiments. Thermal convective velocity is thus more delayed respect to solute transport. The
thermal dispersion mechanism dominates heat propagation in the fractured medium in the carried
out experiments and thus cannot be neglected.
For mass transport the presence of the secondary path and the nonlinear flow regime are the main
factors affecting non – Fickian behavior observed in experimental BTCs, whereas for heat transport
the non - Fickian nature of the experimental BTCs is governed mainly by the heat exchange
mechanism between the fracture network and the surrounding matrix. The presence of a nonlinear
flow regime gives rise to a weak growth on heat transfer phenomena.
Section 5 reports some practical applications of the knowledges acquired from this study on the
convective heat transport in fractured media for exploiting heat recovery and heat dissipation.
Furthermore the estimation of the average effective thermal conductivity suggests that there is a
solid thermal resistance in the fluid to solid heat transfer processes due to the rock – fracture size
ratio. This result matches previous analyses (Pastore et al., 2015) in which a lower heat dissipation
respect to the Tang's solution in correspondence of the single fracture surrounded by a matrix with
more limited heat capacity has been found.
**2 Theoretical background**
**2.1 Nonlinear flow**
With few exceptions, any fracture can be envisioned as two rough surfaces in contact. In cross
section the solid areas representing asperities might be considered as the grains of porous media.
Therefore, in most studies examining hydrodynamic processes in fractured media, the general
equations describing flow and transport in porous media are applied, such as Darcy's law, that
depicts a linear relationship between the pressure gradient and fluid velocity (Whitaker, 1986;
Cherubini and Pastore, 2010)
However, this linearity has been demonstrated to be valid at low flow regimes (Re < 1). For Re > 1
a nonlinear flow behavior is likely to occur (Cherubini, 2013d).
When $Re \gg 1$, a strong inertial regime develops, that can be described by the Forchheimer equation
(Forchheimer, 1901):
$$-\frac{dp}{dx} = \frac{\mu}{k} \cdot u_f + \rho\beta \cdot u_f^2 \tag{1}$$

Where $x$ (m) is the coordinate parallel to the axis of the single fracture ($SF$), $p$ ($ML^{-1}T^{-2}$) is the flow
pressure, $\mu$ ($ML^{-1}T^{-1}$) is the dynamic viscosity, $k$ ($L^2$) is the permeability, $u_f$ ($LT^{-1}$) is the convective
velocity, $\rho$ ($ML^{-3}$) is the density and $\beta$ ($L^{-1}$) is called the inertial resistance coefficient, or non –
Darcy coefficient.
It is possible to express Forchheimer law in terms of hydraulic head $h$ (L):
$$-\frac{dh}{dx} = a' \cdot u_f + b' \cdot u_f^2 \tag{2}$$

The coefficients $a'$ ($TL^{-1}$) and $b'$ ($TL^{-2}$) represent the linear and inertial coefficient respectively
equal to:
$$a' = \frac{\mu}{\rho g k}; \ b' = \frac{\beta}{g} \tag{3}$$

The relationship between hydraulic head gradient and flow rate $Q$ ($L^3T^{-1}$) can be written as:
$$-\frac{dh}{dx} = a \cdot Q + b \cdot Q^2 \tag{4}$$

The coefficients $a$ ($TL^{-3}$) and $b$ ($T^2L^{-6}$) can be related to $a'$ and $b'$:
$$a = \frac{a'}{\omega_{eq}}; \ b = \frac{b'}{\omega_{eq}^2} \tag{5}$$

Where $\omega_{eq}$ (L$^2$) is the equivalent cross sectional area of *SF*.

## 2.2 Heat transfer by water flow in single fractures

Fluid flow and heat transfer in a single fracture (*SF*) undergo advective, diffusive and dispersive
phenomena. Dispersion is caused by small scale fracture aperture variations. Flow channeling is one
example of macrodispersion caused by preferred flow paths, in that mass and heat tend to migrate
through the portions of a fracture with the largest apertures.
In fractured media another process is represented by diffusion into surrounding rock matrix. Matrix
diffusion attenuates the mass and heat propagation in the fractures.
According to the boundary – layer theory (Fahien, 1983), solute mass transfer $q_m$ (ML$^{-2}$) per unit
area at the fracture-matrix interface (Wu et al., 2010) is given by:
$$q_M = \frac{D_m}{\delta}\left(c_f - c_m\right) \tag{6}$$
Where $c_f$ (ML$^{-3}$) is the concentration across fractures, $c_m$ (ML$^{-3}$) is the concentration of the matrix
block surfaces, $D_m$ (LT$^{-2}$) is the molecular diffusion coefficient, and $\delta$ (m) is the thickness of
boundary layer (Wu et al., 2010). For small fractures, $\delta$ may become the aperture $w_f$ (m) of the *SF*.
In analogous manner the specific heat transfer flux $q_H$ (MT$^{-3}$) at the fracture – matrix interface is
given by:
$$q_H = \frac{k_m}{\delta}\left(T_f - T_m\right) \tag{7}$$
Where $T_f$ (K) is the temperature across fractures, $T_m$ (K) is the temperature of the matrix block
surfaces, $k_m$ (MLT$^{-3}$K$^{-1}$) is the thermal conductivity.
The continuity conditions at the fracture – matrix interface requires a balance between mass transfer
rate and mass diffused into the matrix described as:
$$q_M = - D_e \left.\frac{\partial c_m}{\partial z}\right|_{z=w_f/2} \tag{8}$$
Where $z$ (m) is the coordinate perpendicular to the fracture axis and $w_f$ is the aperture of the
fracture.
In the same way the specific heat flux must be balanced by heat diffused into the matrix described
as:
$$q_H = -k_e \frac{\partial T_m}{\partial z}\bigg|_{z=w_f/2} \tag{9}$$
The effective diffusion coefficient takes into account the fact that diffusion can only take place
through pore and fracture openings because mineral grains block many of the possible pathways.
The effective thermal conductivity of a formation consisting of multiple components depends on the
geometrical configuration of the components as well as on the thermal conductivity of each.
The effective terms ($D_e$ instead of $D_m$ and $k_e$ instead of $k_m$) have been introduced in order to include
the effect of various system parameters such as fluid velocity, porosity, surface area, roughness, that
may enhance mass and heat transfer effect. For instance, when large flow velocity occurs,
convective transport is stronger along the centre of the fracture, enhancing the concentration or
temperature gradient at the fracture matrix interface. As known roughness plays an important role in
increasing mass or heat transfer because of increasing turbulent flow conditions.
According to Bodin (2007) the governing equation for the one dimensional advective - dispersive
transport along the axis of a semi-infinite fracture with one – dimensional diffusion in the rock
matrix, in perpendicular direction to the axis of the fracture is:
$$\frac{\partial c_f}{\partial t} + u_f \frac{\partial c_f}{\partial x} = \frac{\partial}{\partial x}\left( D_f \frac{\partial c_f}{\partial x} \right) - \frac{D_e}{\delta} \frac{\partial c_m}{\partial z}\bigg|_{z=w_f/2} \tag{10}$$
Where $D_f$ ($L^2T^{-1}$) is the dispersion. The latter mainly depends on two processes: Aris – Taylor
dispersion and geometrical dispersion. Previous experiments (Cherubini et al., 2012, 2013a, 2013b,
2013c and 2014) show that, due to the complex geometrical and topological characteristics of the
fracture network that create tortuous flow paths, Aris – Taylor dispersion may not develop. A linear
relationship has been found between velocity and dispersion so geometrical dispersion is mostly
responsible for the mixing process along the fracture:
$$D_f = \alpha_{LM} u_f \tag{11}$$
Where $\alpha_{LM}$ (L) is the dispersivity coefficient for mass transport.
Assuming that fluid flow velocity in the surrounding rock matrix is equal to zero, the equation for
the conservation of heat in the matrix is given by:
$$\frac{\partial c_m}{\partial t} = D_a \frac{\partial^2 c_m}{\partial z^2}$$ (12)
Where $D_a$ is the apparent diffusion coefficient of the solute in the matrix expressed as function of
the matrix porosity $\theta_m$, $D_a = D_e / \theta_m$ (Bodin et al., 2007).
Tang et al. (1981) presented an analytical solution for solute transport in semi – infinite single
fracture embedded in a porous rock matrix with a constant concentration at the fracture inlet ($x = 0$)
equal to $c_0$ (ML$^{-3}$) and with an initial concentration equal to zero. The solute concentration in the
fracture $\overline{c}_f$ and in the matrix $\overline{c}_m$ has been given as function of time in Laplace space as follows:
$$\overline{c}_f = \frac{c_0}{s} \exp(vL) \exp\left[ -vL \left\{ 1 + \beta^2 \left( \frac{s^{1/2}}{A} + s \right) \right\}^{1/2} \right]$$ (13)
$$\overline{c}_m = \overline{c}_f \exp\left[ -Bs^{1/2} \left( z - w_f / 2 \right) \right]$$ (14)
Where $s$ is the integral variable of the Laplace transform, $L$ (L) is the length of *SF*, the $v$, $A$, $\beta^2$ and $B$
coefficients are expressed as follows: $v = \dfrac{u_f}{2D_f}$ (15)
$$A = \frac{\delta}{\sqrt{\theta_m D_e}}$$ (16)
$$\beta^2 = \frac{4D_f}{u_f^2}$$ (17)
$$B = \frac{1}{\sqrt{D_e}}$$ (18)
Whereas the gradient of $\overline{c}_m$ at the interface $z = w_f/2$ is:
$$\left. \frac{d\overline{c}_m}{dx} \right|_{x=w_f/2} = -\overline{c}_f B s^{1/2}$$ (19)
Defined the residence time as the average amount of time that the solute spends in the system, on
the basis of these analytical solutions the probability density function (*PDF*) of the solute residence
time in the single fracture in the Laplace space can be expressed as:
$$\overline{\Gamma}(s) = \exp(vL)\exp\left[-vL\left\{1 + \beta^2\left(\frac{s^{1/2}}{A} + s\right)\right\}^{1/2}\right]$$ (20)

Assuming that density and heat capacity are constant in time, the heat transport conservation equation in $SF$ can be expressed as follows:

$$\frac{\partial T_f}{\partial t} + u_f\frac{\partial T_f}{\partial x} = \frac{\partial}{\partial x}\left(D_{fH}\frac{\partial T_f}{\partial x}\right) - \frac{k_e}{\rho_w C_w \delta}\frac{\partial T_m}{\partial z}\bigg|_{z=w_f/2}$$ (21)

Where $\rho_w$ (ML$^{-3}$), $C_w$ (L$^2$T$^2$K$^{-1}$) represent the density, the specific heat capacity of the fluid within $SF$ respectively. $D_f$ for heat transport assumes the following expression:

$$D_{fH} = \frac{\lambda_L}{\rho_w C_w}$$ (22)

Where $\lambda_L$ is the thermodynamic dispersion coefficient (MLT$^{-3}$K$^{-1}$). Sauty et al. (1982) and de Marsily (1986) proposed an expression for the thermal dispersion coefficient where the thermal dispersion term varies linearly with velocity and depends on the heterogeneity of the medium, as for solute transport:

$$\lambda_L = k_0 + \alpha_{LH}\rho_w C_w u_f$$ (23)

Where $k_0$ is the bulk thermal conductivity (MLT$^{-3}$K$^{-1}$) and $\alpha_{LH}$ (L) is the longitudinal thermal dispersivity.

The heat transport conservation equation in the matrix is expressed as follows:

$$\rho_m C_m\frac{\partial T_m}{\partial t} = k_e\frac{\partial^2 T_m}{\partial z^2}$$ (24)

Note that the governing equations of heat and mass transport highlight similarities between the two processes, thus Tang's solution can be used also for heat transport.

In terms of heat transport, the coefficients $v$, $A$, $\beta^2$ and $B$ are expressed as follows:

$$v = \frac{u_f}{2D_{fH}}$$ (25)
$$A = \frac{\delta}{\sqrt{\theta D_e}} \tag{26}$$
where $\theta = \rho_m C_m / \rho_w C_w$ and $D_e = k_e / \rho_w C_w$.
$$\beta^2 = \frac{4D_f}{u_f^2} \tag{27}$$
$$B = \frac{1}{\sqrt{D_e}} \tag{28}$$
Three characteristic time scales can be defined:
$$t_u = \frac{L}{u_f}; \quad t_d = \frac{L^2}{D_f}; \quad t_e = \frac{\delta^2}{D_e} \tag{29}$$
Where $L$ (L) is the characteristic length, $t_u$ (T), $t_d$ (T) and $t_e$ (T) represent the characteristics time
scales of convective transport, dispersive transport and loss of the mass or heat into the surrounding
matrix.
The relative effect of dispersion, convection and matrix diffusion on mass or heat propagation in the
fracture can be evaluated by comparing the corresponding time scale.
Peclet number $P_e$ is defined as the ratio between dispersive ($t_d$) to convective ($t_u$) transport times:
$$Pe = \frac{t_d}{t_u} = \frac{u_f L}{D_f} \tag{30}$$
At high Peclet numbers transport processes are mainly governed by convection, whereas at low
Peclet numbers it is mainly dispersion that dominates.
Another useful dimensionless number, generally applied in chemical engineering, is the Damköhler
number that can be used in order to evaluate the influence of matrix diffusion on convection
phenomena. $Da$ relates the convection time scale to the exchange time scale.
$$Da = \frac{t_u}{t_e} = \frac{\alpha L}{u_f} \tag{31}$$
Where $\alpha$ (T$^{-1}$) is the exchange rate coefficient corresponding to:
$$\alpha = \frac{D_e}{\delta^2}$$ (32)
Note that the inverse of $t_e$ has the same meaning of the exchange rate coefficient $\alpha$ ($T^{-1}$).
When $t_e$ values are of the same order of magnitude as the transport time $t_u$ ($Da \cong 1$), diffusive
processes in the matrix are more relevant. In this case concentration or temperature distribution
profiles are characterized by a long tail.
When $t_e \gg t_u$ ($Da \ll 1$) the fracture – matrix exchange is very slow and it does not influence mass
or heat propagation. On the contrary when $t_e \ll t_u$ ($Da \gg 1$) the fracture matrix exchange is rapid,
there is instantaneous equilibrium between fracture and matrix and they have the same
concentration or temperature. These two circumstances close the standard advective – dispersive
transport equation.
The product between $Pe$ and $Da$ represents another dimensionless group which is a measure of
transport processes:
$$Pe \times Da = \frac{t_d}{t_e} = \frac{\alpha L^2}{D_f}$$ (33)
When $Pe \times Da$ increases $t_e$ decreases more rapidly than $t_d$, and subsequently the mass or heat
diffusion into the matrix may be dominant on the longitudinal dispersion.
**2.3 Explicit network model (ENM)**
The 2D Explicit Network Model (ENM) depicts the fractures as 1D pipe elements forming a 2D –
pipe network and therefore expressly takes the fracture network geometry into account. The ENM
model permits to understand the physical meaning of flow and transport phenomena and therefore
to obtain a more accurate estimation of flow and transport parameters.
With the assumption that a $j^{th}$ $SF$ can be schematized by a 1D – pipe element, the Forchheimer
model can be used to write the relationship between head loss $\Delta h_j$ (L) and flow rate $Q_j$ ($L^3 T^{-1}$) in
finite terms:
$$\frac{\Delta h_j}{L_j} = aQ_j + bQ_j^2 \Rightarrow \Delta h_j = \left[ L_j \left( a + bQ_j \right) \right] Q_j$$ (34)
Where $L_j$ (L) is the length of $j^{\text{th}}$ SF, $a$ (TL$^{-3}$) and $b$ (T$^2$L$^{-6}$) represent the Forchheimer parameters
written in finite terms. The term in the square brackets constitutes the resistance to flow $R_j\left(Q_j\right)$
(TL$^{-2}$) of $j^{\text{th}}$ SF .
In case of steady – state conditions and for a simple 2D fracture network geometry, a
straightforward manner can be applied to obtain the solution of flow field by applying the first and
second Kirchhoff's laws.
In a 2D fracture network, fractures can be arranged in series and/or in parallel. Specifically, in a
network in which fractures are set in a chain, the total resistance to flow is calculated by simply
adding up the resistance values of each single fracture. The flow in a parallel fracture network
breaks up, with some flowing along each parallel branch and re – combining when the branches
meet again. In order to estimate the total resistance to flow the reciprocals of the resistance values
have to be added up and then the reciprocal of the total has to be calculated. The flow rate $Q_j$ across
the generic fracture $j$ of the parallel network can be calculated as (Cherubini et al., 2014):
$$Q_j = \sum_{i=1}^{n} Q_i \left[ \frac{1}{R_j} \left( \sum_{i=1}^{n} \frac{1}{R_i} \right)^{-1} \right] \tag{35}$$
Where $\displaystyle\sum_{i=1}^{n} Q_i$ ( LT$^{-3}$) is the sum of the mass flow rates at fracture intersections in correspondence of
the inlet bond of $j$ fracture, whereas the term in square brackets represents the probability of water
distribution of $j$ fracture $P_{Q,j}$.
Once known the flow field in the fracture network, to obtain the PDF at a generic node the PDFs of
each elementary path that reaches the node have to be summed up. They can be calculated as the
convolution product of the PDFs of each single fracture composing the elementary path.
Definitely the BTC describing the concentration in the fracture as function of time at the generic
node, using the convolution theorem, can be obtained as follows:
$$c_f\left(t\right) = c_0 + c_{inj}\left(t\right) * L^{-1}\left[ \sum_{i=1}^{N_p} \prod_{j=1}^{n_{f,i}} P_{M,j} \overline{\Gamma}_j\left(s\right) \right] \tag{36}$$
Where $c_0$ (ML$^{-3}$) is the initial concentration and $c_{inj}$ (ML$^{-3}$) is the concentration injection function, $*$
is the convolution operator, $L^{-1}$ represents the inverse Laplace transform operator, $N_p$ is the number
of the paths reaching the node, $n_{f,i}$ is the number of the SF belonging to the elementary path $i^{\text{th}}$, $P_{M,j}$
and $\overline{\Gamma}(s)$ are the mass distribution probability and the *PDF* in the Laplace space of the generic $j^{th}$
*SF* respectively. Inverse Laplace transform $L^{-1}$ can be solved numerically using Abate et al. (2006)
algorithm.
At the same way the BTC $T_f$ which describes the temperature in the fracture as function of time at
the generic node can be written as:
$$T_f(t) = T_0 + T_{inj}(t) * L^{-1}\left[ \sum_{i=1}^{N_p} \prod_{j=1}^{n_{f,i}} P_{H,j} \overline{\Gamma}_j(s) \right]$$ (37)
Where $T_0$ (K) is the initial temperature, $T_{inj}$ (K) is the temperature injection function and $P_{H,j}$ is the
heat distribution probability.
$P_{M,j}$ and $P_{H,j}$ can be estimated as the probabilities of the mass and heat distribution at the inlet bond
of each individual *SF* respectively. The mass and heat distribution is proportional to the
correspondent flow rates:
$$P_{M,j} = P_{H,j} = \frac{Q_j}{\sum_{i=1}^{n} Q_i}$$ (38)
Note that if Equation 38 is valid, the probability of water distribution is equal to the probabilities of
mass and heat distribution (term in square brackets in Equation 34). Definitely the ENM model
regarding each *SF* can be described by four parameters ($u_{f,j}$, $D_{f,j}$, $\alpha_j$, $P_{Q,j}$).

## 3 Material and methods

### 3.1 Description of the experimental apparatus

The heat transfer tests have been carried out on the experimental apparatus previously employed to
perform flow and tracer transport experiments at bench scale (Cherubini et al. 2012, 2013a, 2013b,
2013c and 2014). However, the apparatus has been modified in order to analyze heat transport
dynamics. Two thermocouples have been placed at the inlet and the outlet of a selected fracture
path of the limestone block with parallelepiped shape ($0.6 \times 0.4 \times 0.08$ m$^3$) described in previous
studies. A TC – 08 Thermocouple Data Logger (pico Technology) with a sampling rate of 1 second
has been connected to the thermocouples. An extruded polystyrene panel with thermal conductivity
equal to 0.034 Wm$^{-1}$K$^{-1}$ and thickness 0.05 m has been used to thermally insulate the limestone
block which has then been connected to a hydraulic circuit. The head loss between the upstream
tank connected to the inlet port and the downstream tank connected to the outlet port drives flow of
water through the fractured block. An ultrasonic velocimeter (DOP3000 by Signal Processing) has
been adopted to measure the instantaneous flow rate that flows across the block. An electric boiler
with a volume of $10^{-2}$ m$^3$ has been used to heat the water. In a flow cell located in correspondence
of the outlet port a multiparametric probe is positioned for the instantaneous measurement of
pressure (dbar), temperature (°C) and electric conductivity ($\mu$S cm$^{-1}$). Figure 1a shows the fractured
block sealed with epoxy resin, Figure 1b shows the thermal insulated fractured block connected to
the hydraulic circuit, whereas the schematic diagram of the experimental apparatus is shown in
Figure 2.
**3.2 Flow experiments.**
The average flow rate through the selected path can be evaluated as:
$$\bar{Q} = \frac{S_1}{t_1 - t_0}\left(h_1 - h_0\right) \tag{39}$$

Where $S_1$ (L$^2$) is the cross section area of the flow cell, $\Delta t = t_1 - t_0$ is the time for the flow cell to be
filled from $h_0$ (L) and $h_1$ (L). To calculate the head loss between the upstream tank and the flow cell
the following expression is adopted:
$$\Delta h = h_c - \frac{h_0 + h_1}{2} \tag{40}$$

Where $h_c$ is the hydraulic head measured in the upstream tank. Several tests have been carried out
varying the control head, and in correspondence of each value of the average flow rate and head
loss the average resistance to flow has been determined as:
$$\bar{R}\left(\bar{Q}\right) = \left[\frac{S_1}{t_1 - t_0}\ln\left(\frac{h_0 - h_c}{h_1 - h_c}\right)\right]^{-1} \tag{41}$$

**3.3 Solute and temperature tracer tests**
Solute and temperature tracer tests have been conducted through the following steps.
As initial condition, a specific value of hydraulic head difference between the upstream tank and
downstream tank has been assigned. At $t = 0$ the valve $a$ is closed so as the hydrostatic head inside
the block assumes the same value to the one in the downstream tank. At $t = 10$ s the valve $a$ is
opened.
For solute tracer test at time $t = 60$ s by means of a syringe, a mass of $5 \times 10^{-4}$ kg sodium chloride is
injected into the inlet port. Due to the very short source release time, the instantaneous source
assumption can be adopted which assumes the source of solute as an instantaneous injection (pulse).
The multiparametric probe located within the flow cell measures the solute BTC.
As concerns thermal tracer tests at the time $t = 60$ s the valve $d$ is opened while the valve c is
closed. In such a way a step temperature function in correspondence of the inlet port $T_{inj}(t)$ is
imposed and measured by the first thermocouple. The other thermocouple located inside the outlet
port is used to measure the thermal BTC.
The ultrasonic velocimeter is used in order to measure the instantaneous flow rate, whereas a
multiparametric probe located at the outlet port measures the pressure and the electric conductivity.
**4 Results and discussion**
**4.1 Flow characteristics**
The Kirchhoff laws have been used in order to estimate the flow rates flowing in each single
fracture. In Figure 3 a sketch of the 2D pipe conceptualization of the fracture network is reported.
The resistance to flow of each *SF* can be evaluated as the square bracket in Equation (34). For
simplicity the linear and non linear terms have been considered constant and equal for each *SF*.
The resistance to flow for the whole fracture network $\bar{R}(\bar{Q})$ can be evaluated as the sum of the
resistance to flow of each *SF* arranged in chain and the total resistance of the parallel branches
equal to the reciprocal of the sum of the reciprocal of the resistance to flow of each parallel branch:
$$
\bar{R}(\bar{Q}) = R_1(Q_0) + R_2(Q_0) + \left( \frac{1}{R_6(Q_1)} + \frac{1}{R_3(Q_2) + R_4(Q_2) + R_5(Q_2)} \right)^{-1} +
$$
$$
+ R_7(Q_0) + R_8(Q_0) + R_9(Q_0)
$$
(42)

Where $R_j$ with j = 1 – 9 represents the resistance to flow of each *SF*, $Q_0$ is the injection flow rate,
$Q_1$ and $Q_2$ are the flow rates flowing in the parallel branch 6 and 3 – 4 – 5 respectively.
The flow rate $Q_1$ is determined in iterative manner using the following iterative equation derived by
the Equation (35) at the node 3:
$$Q_1^{k+1} = Q_0 \left[ \frac{1}{R_6\left(Q_1^k\right)} \left( \frac{1}{R_3\left(Q_0 - Q_1^k\right) + R_4\left(Q_0 - Q_1^k\right) + R_5\left(Q_0 - Q_1^k\right)} + \frac{1}{R_6\left(Q_1^k\right)} \right)^{-1} \right]$$ (43)
Whereas the flow rate $Q_2$ is determined merely as:
$$Q_2 = Q_0 - Q_1$$ (44)
The linear and nonlinear terms representative of the whole fracture network have been estimated
matching the average experimental resistance to flow resulting from Equation (41) with resistance
to flow estimated from Equation (42).
The linear and nonlinear term are equal respectively to $a = 7.345 \times 10^4$ sm$^{-3}$ and $b = 11.65 \times 10^9$ s$^2$m$^{-6}$.
Inertial forces dominate viscous ones when the Forchheimer number ($Fo$) is higher than one. $Fo$ can
be evaluated as the ratio between the non linear loss $\left(bQ^2\right)$ and the linear loss $\left(aQ\right)$. The critical
flow rate $Q_{crit}$ which represents the value of flow rate for which $Fo = 1$ is derived as the ratio
between $a$ and $b$ resulting $Q_{crit} = 6.30 \times 10^{-6}$ m$^3$s$^{-1}$.
Because of the nonlinearity of flow, varying the inlet flow rate $Q_0$ the ratio between the flow rates
$Q_1$ and $Q_2$ flowing respectively in the branches 6 and 3 – 5 is not constant. When $Q_0$ increases $Q_2$
increases faster than $Q_1$. The probability of water distribution of the branch 6 $P_{Q,6}$ is evaluated as
the ratio between $Q_0$ and $Q_1$, whereas the probability of water distribution of the branch 3 – 5 is
equal to $P_{Q,3-5} = 1 - P_{Q,6}$.
**4.2 Fitting of breakthrough curves and interpretation of estimated model parameters**
The behavior of mass and heat transport has been compared varying the injection flow rates. In
particular 21 tests in the range $1.83 \times 10^{-6}$ - $1.26 \times 10^{-5}$ m$^3$s$^{-1}$ (Re in the range $17.5 – 78.71$) for heat
transport have been made and compared with the 55 tests in the range $1.32 \times 10^{-6}$ - $8.34 \times 10^{-6}$ m$^3$s$^{-1}$
(Re in the range $8.2 – 52.1$) for solute transport presented in previous studies.
The observed heat and mass BTCs for different flow rates have been individually fitted using the
ENM approach presented in section 2.3. For simplicity the transport parameters $u_f$, $D_f$ and $\alpha$ are
assumed equal for all branches of the fracture network. The probability of mass and heat
distribution are assumed equal to the probability of water distribution.

The experimental BTCs are fitted using Equation (36) and Equation (37) for mass and heat transport respectively. Note that for mass transport $c_{inj}(t)$ supposing the instantaneous injection condition becomes a Dirac delta function.

The determination coefficient ($r^2$) and the root mean square error ($RMSE$) have been used in order to evaluate the goodness of fit.

Tables 1 and 2 show the values of transport parameters, the $RMSE$ and $r^2$ for mass and heat transport respectively. Furthermore Figure 4 and Figure 5 show the fitting results of BTCs for some values of $Q_0$.

The results presented in Tables 1 and 2 highlight that: the estimated convective velocities $u_f$ for heat transport are lower than for mass transport. Whereas the estimated dispersion $D_f$ for heat transport is higher than for mass transport. Regarding the transfer rate coefficient $\alpha$, it assumes very low values for mass transport relatively to the convective velocity. Instead for heat transport the exchange rate coefficient is of the same order of magnitude of the convective velocity and, considering a characteristic length equal to $L = 0.601$ m corresponding to the length of the main path of the fracture network, the effect of dual – porosity is very strong and cannot be neglected relatively to the investigated injection flow range. Both mass and heat transport show a satisfactory fitting. In particular manner, $RMSE$ varies in the range 0.0015 – 0.0180 for mass transport and in the range 0.0030 – 0.236 for heat transport, whereas $r^2$ varies in the range 0.9863 – 0.9987 for mass transport and in the range 0.0963 – 0.9998 for heat transport.

In order to investigate the different behavior between mass and heat transport, the relationships between injection flow rate and the transport parameters have been analyzed. In Figure 6 the relationship between $u_f$ and $Q_0$ is reported. Whereas in Figures 7 and 8 are reported the dispersion coefficient $D_f$ and the exchange term $\alpha$ as function of $u_f$ respectively. The figures show a very different behavior between mass and heat transport.

Regarding mass transport experiments according to previous studies (Cherubini at al., 2013a, 2013b, 2013c and 2014) the figure 5 shows that for values of $Q_0$ higher than $4 \times 10^{-6}$ m$^3$s$^{-1}$ $u_f$ increases less rapidly. This behavior was due to the presence of inertial forces that gave rise to a retardation effect on solute transport.

Instead Figure 7 shows a linear relationship between $u_f$ and $D_f$ suggesting that inertial forces did not exert any effect on dispersion and that geometrical dispersion dominates the Aris – Taylor dispersion.

In the same way as for mass transport, for heat transfer a linear relationship is evident between
dispersion and convective velocity. Even if heat convective velocity is lower than solute advective
velocity, the longitudinal thermal dispersivity assumes higher values than the longitudinal solute
dispersivity. Also for heat transport experiments a linear relationship between $u_f$ and $D_f$ has been
found.
Figure 8 shows the exchange rate coefficient α as function of the convective velocity $u_f$ for both
mass and heat transport.
Regarding the mass transport, the estimated exchange rate coefficient $\alpha$ is much lower than the
convective velocity. These results suggest that in the case study fracture – matrix exchange is very
slow and it may not influence mass transport. Non Fickian behavior observed in the experimental
BTCs is therefore dominated mainly by the presence of inertial forces and the parallel branches.
A very different behavior is observed for heat transport. Heat convective velocity does not seem to
be influenced by the presence of the inertial force whereas $u_f$ is influenced by fracture matrix
exchange phenomena resulting in a significant retardation effect. Once the model parameters for
each flow rate have been determined, the unit response function ($f_{URF}$), corresponding to the *PDF*
obtained from impulsive injection of both solute and temperature tracers, is obtained. The unit
response function can be characterized using the time moments and tail character analysis.
The mean residence time $t_m$ assumes the following expression:
$$t_m = \frac{\int_0^\infty t f_{URF}(t)\,dt}{\int_0^\infty f_{URF}(t)\,dt}$$   (45)
Whereas the n$^{th}$ normalized central moment of distribution of the $f_{URF}$ versus time can be written as:
$$\mu_n = \frac{\int_0^\infty (t - t_m)^n f_{URF}(t)\,dt}{\int_0^\infty f_{URF}(t)\,dt}$$   (46)
The second moment $\mu_2$ can be used in order to evaluate the dispersion relative to $t_m$, whereas the
skewness is a measure of the degree of asymmetry and it is defined as follows:
$$S = \mu_3 / \mu_2^{3/2}$$   (47)
The tailing character $t_c$ can be described as:
$$t_c = \frac{\Delta t_{fall}}{\Delta t_{rise}}$$ (48)
Where $\Delta t_{fall}$ denotes the duration of the falling limb defined as the time interval from the peak to the
tail cutoff which is the time when the falling limb first reaches a value that is 0.05 times the peak
value. $\Delta t_{rise}$ is defined as the time interval from the first arrival to the peak. This quantity provides a
measure of the asymmetry between the rising and falling limbs. A value of $t_c$ significantly higher
than 1 indicates an elongated tail compared to the rising limb (Cherubini et al., 2010).
In Figure 9 is reported the residence time versus the injection flow rates. The figure highlights that
$t_m$ for heat transport is about 3 times higher than for mass transport. In particular way $t_m$ varies in
the range 40. 3 - 237.1 $s$ for mass transport and in the range 147.8 – 506.9 $s$ for heat transport. This
result still highlights that heat transport is more delayed than mass transport.
In same way the skewness $S$ (Figure 10) and tailing character $t_c$ (Figure 11) are reported as function
of $Q_0$.
A different behavior for heat and mass transport is observed for the skewness coefficient. For heat
transfer the skewness shows a growth trend which seems to decrease after $Q_0 = 3{\times}10^{-6}$ m$^3$s$^{-1}$. Its
mean value is equal to 2.714. For solute transport the S does not show a trend, and assumes a mean
value equal to 2.018.
The tailing character does not exhibit a trend for both mass and heat transport. In either cases $t_c$ is
significantly higher than 1, specifically 7.70 and 30.99 for mass and heat transport respectively.
In order to explain the transport dynamics, the trends of dimensionless numbers $Pe$ and $Da$ varying
the injection flow rate have been investigated. The Figure 12 shows the $Pe$ as function of $Q_0$ for
both mass and heat experiments. As concerns mass experiments $Pe$ increases as $Q_0$ increases,
assuming a constant value for high values ($Pe = 7.5$) of $Q_0$. For heat transport a different behavior is
observed, $P_e$ showing a constant trend and being always lower than one. Even if the injection flow
rate is relatively high, thermal dispersion is the dominating mechanism in heat transfer.
Figure 12 reports $Da$ as function of $Q_0$. For mass transport $Da$ assumes very low values, of the
order of magnitude of $10^{-4}$.
The convective transport scale is very low respect to the exchange transport scale, thus the mass
transport in each single fracture can be represented with the classical advection dispersion model.
As regards heat transport $Da$ assumes values around the unit showing a downward trend as injection
flow rate increases switching from higher to lower values than the unit. As injection flow rate
increases the convective transport time scale reduces more rapidly than the exchange time scale.
These arguments can be explained because the relationships between $Q_0$ and $u_f$ show a change of
slope when $Da$ becomes lower than the unit. In other words when $Da$ is higher than the unit the
exchange between fracture and matrix dominates on the convective transport giving rise to a more
enhanced delay on heat transport, conversely when $Da$ is lower than one convective transport
dominates on fracture- matrix interactions and the delay effect is reduced.
Furthermore this effect is evident also on the trend observed in the graph $S - Q_0$ (Figure 10). For
values of $Da$ lower than the unit a change of slope is evident, the skewness coefficient increases
more slowly. Thus for $Da>1$ the early arrival and the tail effect of $BTC$ increase more rapidly than
for $Da<1$.
Note that even if $Da$ presents a downward trend as $Q_0$ increases, when the latter exceeds $Q_{crit}$ a
weak growth trend for $Da$ is detected, that however assumes values lower than the unit.
The Figure 14 shows the dimensionless group $Pe \times Da$ varying the injection flow rate. Regarding
mass transport $Pe \times Da$ is of the order of magnitude of $10^{-3}$ confirming the fact that the fracture –
matrix interaction can be neglected relatively to the investigated range of injection flow rates. For
heat transport $Pe \times Da$ assumes values just below the unit, with a downward trend as $Q_0$ increases. $t_d$
and $t_e$ have the same order of magnitude.
In order to find the optimal conditions for heat transfer in the analyzed fractured medium the
thermal power exchanged per unit temperature difference $\dot{Q}$ ($ML^2T^{-1}K^{-1}$) for each injection flow
rate in quasi steady state conditions can be estimated. The thermal power exchanged can be written
as:
$$\dot{Q} = \rho C_w Q_0 \left( T_{inj} - T_{out} \right) \tag{49}$$
The outlet temperature $T_{out}$ can be evaluated as function of the $f_{URF}$ using the following expression:
$$T_{out} = T_0 + \left( T_{inj} - T_0 \right) \int_0^\infty f_{URF}(t)\, dt \tag{50}$$
Substituting the Equation (50) in the Equation (49) the thermal power exchanged per unit
temperature difference is:
$$\frac{\dot{Q}}{\left(T_{inj} - T_0\right)} = \left(1 - \int_0^\infty f_{URF}(t)\,dt\right)\rho C_W Q_0 \qquad (51)$$

Figure 15 shows the similarities between the relationship $\dot{Q}/\left(T_{inj} - T_0\right)$ - $Q_0$ (Figure 15a) and $Da - Q_0$
(Figure 14b). Higher $Da$ values correspond to higher values of $\dot{Q}/\left(T_{inj} - T_0\right)$. The thermal power
exchanged increases as the Damköhler number increases as shown in Figure 15c. These results
highlight that for the observed case study the optimal condition for thermal exchange in the
fractured medium is obtained when the exchange time scale is lower than the convective transport
scale or rather when the dynamics of fracture – matrix exchange are dominant on the convective
ones.
Moreover in a similar way to $Da$, $\dot{Q}/\left(T_{inj} - T_0\right)$ shows a weak growth trend when $Q_0$ exceeds $Q_{crit}$.
This means that the nonlinear flow regime improves the fracture – matrix thermal exchange,
however at high values of injection flow rates convective and dispersion time scales are less than
the exchange time scale. Nevertheless these results have been observed in a small range of $Da$
numbers close to the unit. In order to generalize these results a larger range of $Da$ numbers should
be investigated.
In order to estimate the effective thermal conductivity coefficient $k_e$, the principle of conservation of
heat energy can be applied to the whole fractured medium. Neglecting the heat stored in the
fractures, the difference between the heat measured at the inlet and at the outlet must be equal to the
heat diffused into the matrix:
$$\rho C_W Q_0 \left(T_{inj} - T_{out}\right) = \int_{A_f} k_e \left.\frac{dT_m}{dz}\right|_{z=wf/2} dA_f \qquad (52)$$

where $A_f$ is the whole surface area of the whole active fracture network and the gradient of $T_m$ can
be evaluated according to Equation (19) using temperature instead of concentration as variable.
Then the average effective thermal conductivity $\overline{k_e}$ can be obtained as:
$$\overline{k_e} = \frac{\rho_w C_w Q_0 \left(T_{inj} - T_{out}\right)}{\int_{A_f} \left.\frac{dT}{dz}\right|_{z=wf/2} dA_f} \qquad (53)$$

The average effective thermal conductivity has been estimated for each injection flow rate (Figure
16) and assumes a mean value equal to $\bar{k}_e = 0.1183\,\mathrm{Wm^{-1}K^{-1}}$. The estimated $\bar{k}_e$ is one order of
magnitude lower than the thermal conductivity coefficient reported in the literature (Robertson,
1988). Fractured media have a lower capacity for diffusion as opposed to the Tang's model which
has unlimited capacity. There is a solid thermal resistance in the fluid to solid heat transfer
processes which depends on the rock – fracture size ratio.
This result is coherent with previous analyses on heat transfer carried out on the same rock sample
(Pastore et al., 2015). In this study Pastore et al. (2015) found that the ENM model failed to model
the behavior of heat transport in correspondence of parallel branches where the hypothesis of
Tang's solution of single fracture embedded in a porous medium having unlimited capacity cannot
be considered valid. In parallel branches the observed BTCs are characterized by less retardation of
heat propagation as opposed to the simulated BTCs.
**5 Conclusions**
Aquifers offer a possibility of exploiting geothermal energy by withdrawing the heat from
groundwater by means of a heat pump and subsequently supplying the water back into the aquifer
through an injection well. In order to optimize the efficiency of the heat transfer system and
minimize the environmental impacts, it is necessary to study the behavior of convective heat
transport especially in fractured media, where flow and heat transport processes are not well known.
Laboratory experiments on the observation of mass and heat transport in a fractured rock sample
have been carried out in order to analyse the contribution of thermal dispersion in heat propagation
processes, the contribution of nonlinear flow dynamics on the enhancement of thermal matrix
diffusion and finally the optimal heat recovery and heat dissipation strategies.
The parameters that control mass and heat transport have been estimated using the ENM model
based on Tang's solution.
Heat transport shows a very different behavior compared to mass transport. The estimated transport
parameters show differences of several orders of magnitude. Convective thermal velocity is lower
than solute velocity, whereas thermal dispersion is higher than solute dispersion, mass transfer rate
assumes a very low value suggesting that fracture – matrix mass exchange can be neglected. Non -
fickian behavior of observed solute BTCs is mainly due to the presence of the secondary path and
nonlinear flow regime. Contrarily heat transfer rate is comparable with convective thermal velocity
giving rise to a retardation effect on heat propagation in the fracture network.

The discrepancies detected in transport parameters are moreover observable through the time moment and tail character analysis which demonstrate that the dual porosity behavior is more evident in the thermal BTCs than in the solute BTCs.

The dimensionless analysis carried out on the transport parameters proves that as the injection flow rate increases thermal convection time scale decreases more rapidly than the thermal exchange time scale, explaining the reason why the relationship $Q_0 – u_f$ shows a change of slope for $Da$ lower than the unit.

Thermal dispersion dominates heat transport dynamics, the Peclet number and the product between Peclet number and Damköhler number is almost always less than the unit.

The optimal conditions for thermal exchange in a fracture network have been investigated. The power exchanged increases in a potential way as Da increases in the observed range.

The Explicit Network Model is an efficient computation methodology to represent flow, mass and heat transport in fractured media, as 2D and/or 3D problems are reduced to resolve a network of 1D pipe elements. Unfortunately in field case studies it is difficult to obtain the full knowledge of the geometry and parameters such as the orientations and aperture distributions of the fractures needed by the ENM even by means of field investigation methods. However in real case studies the ENM can be coupled with continuum models in order to represent greater discontinuities respect to the scale of study that generally give rise to preferential pathways for flow, mass and heat transport.
A method to represent the topology of the fracture network is represented by multi fractal analisys analysis as discussed in Tijera at al. (2009) and Tarquis at al. 2014.
This study has permitted to detect the key parameters to design devices for heat recovery and heat dissipation that exploit the convective heat transport in fractured media.
Heat storage and transfer in fractured geological systems is affected by the spatial layout of the discontinuities.
Specifically, the rock – fracture size ratio which determines the matrix block size is a crucial element in determining matrix diffusion on fracture – matrix surface.
The estimation of the average effective thermal conductivity coefficient shows that it is not efficient to store thermal energy in rocks with high fracture density because the fractures are surrounded by a matrix with more limited capacity for diffusion giving rise to an increase in solid thermal resistance.
In fact, if the fractures in the reservoir have a high density and are well connected, such that the matrix blocks are small, the optimal conditions for thermal exchange are not reached as the matrix blocks have a limited capability to store heat.

On the other hand, isolated permeable fractures will tend to lead to the more distribution of heat
throughout the matrix.
Therefore, subsurface reservoir formations with large porous matrix blocks will be the optimal
geological formations to be exploited for geothermal power development.
The study could help to improve the efficiency and optimization of industrial and environmental
systems, and may provide a better understanding of geological processes involving transient heat
transfer in the subsurface.
Future developments of the current study will be carrying out investigations and experiments aimed
at further deepening the quantitative understanding of how fracture arrangement and matrix
interactions affect the efficiency of storing and dissipation thermal energy in aquifers. This could be
achieved by means of using different formations with different fracture density and matrix porosity.

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

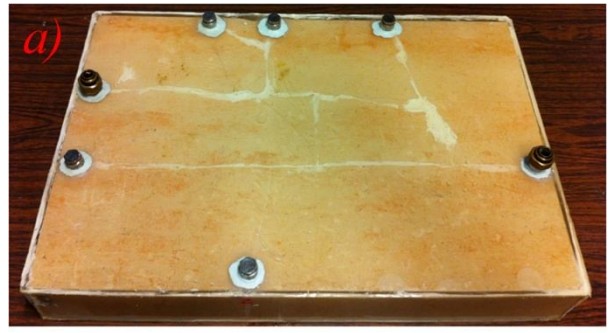

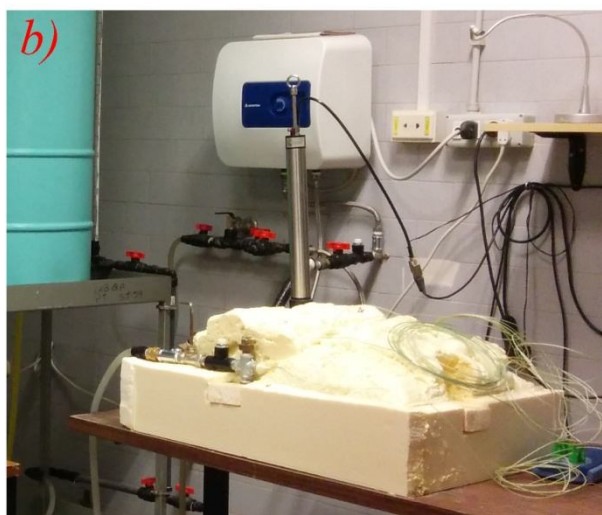


**Figure 1. a) fractured block sealed with epoxy resin. b) thermal insulated fracture block connected to the hydraulic circuit.**

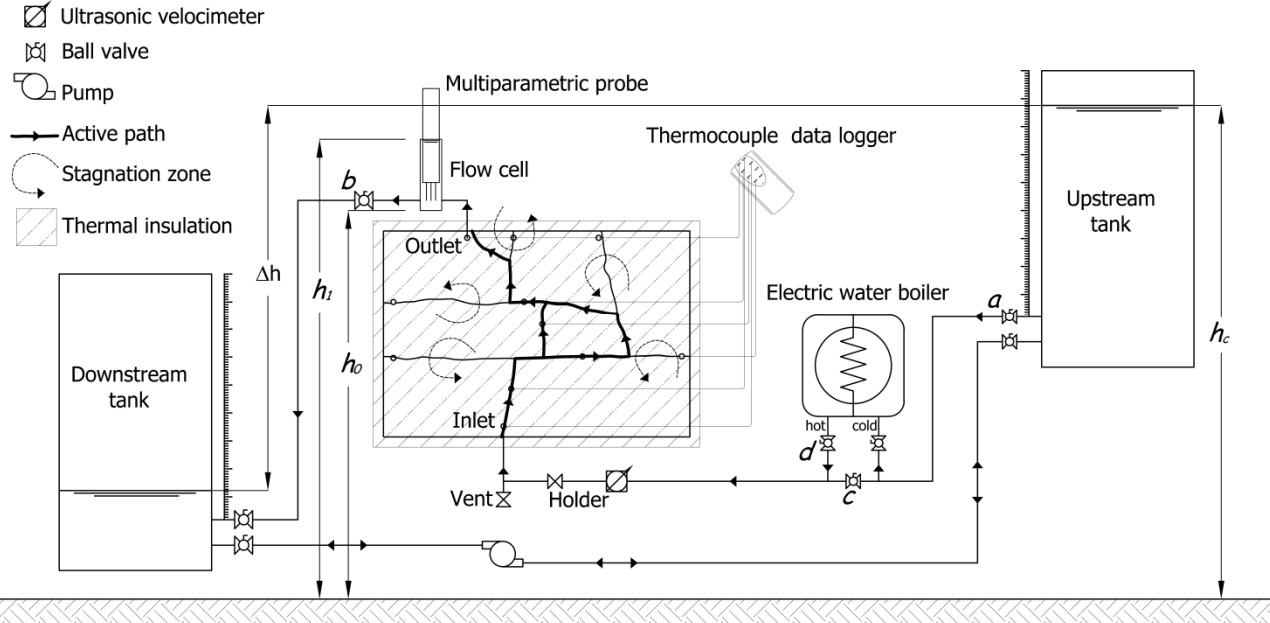


**Figure 2. Schematic diagram of the experimental setup.**

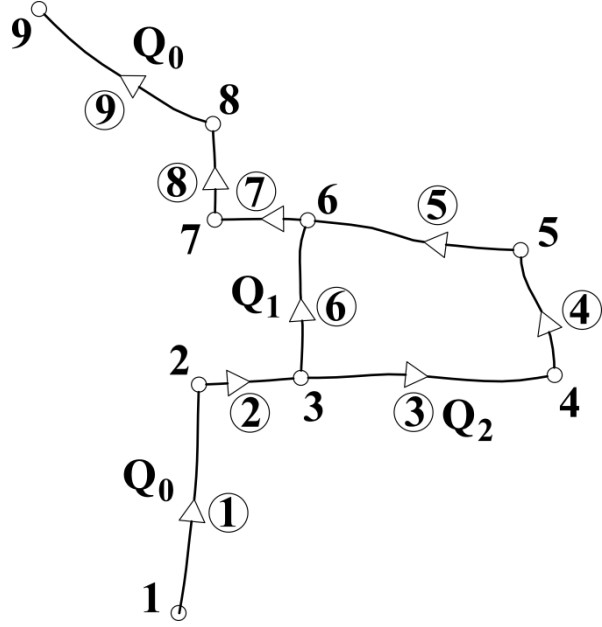

**Figure 3. Two dimensional pipe network conceptualization of the fracture network of the fractured rock block in Figure1. $Q_0$**
**is the injection flow rate, $Q_1$ and $Q_2$ are the flow rates that flowing in the parallel branch 6 and 3-4-5 respectively.**




| Injection flow rate $Q_0$ $(m^3 s^{-1}) \times 10^{-6}$ | Convective velocity $u_f$ $(ms^{-1}) \times 10^{-3}$ | | | Dispersion $D_f$ $(ms^{-2}) \times 10^{-3}$ | | | Exchange rate coefficient $\alpha$ $(s^{-1}) \times 10^{-6}$ | | | RMSE | $r^2$ |
|---|---|---|---|---|---|---|---|---|---|---|---|
| 1.319 | 4.38 | ÷ | 4.47 | 0.68 | ÷ | 0.70 | 4.80 | ÷ | 5.06 | 0.0053 | 0.9863 |
| 1.843 | 6.21 | ÷ | 6.28 | 0.57 | ÷ | 0.58 | 2.86 | ÷ | 3.01 | 0.0026 | 0.9954 |
| 2.234 | 6.54 | ÷ | 6.59 | 0.66 | ÷ | 0.67 | 3.09 | ÷ | 3.13 | 0.0017 | 0.9976 |
| 2.402 | 7.64 | ÷ | 7.68 | 0.67 | ÷ | 0.67 | 2.65 | ÷ | 2.68 | 0.0015 | 0.9983 |
| 2.598 | 9.88 | ÷ | 9.94 | 0.80 | ÷ | 0.82 | 2.76 | ÷ | 2.84 | 0.0015 | 0.9987 |
| 2.731 | 8.27 | ÷ | 8.35 | 0.75 | ÷ | 0.76 | 2.80 | ÷ | 2.91 | 0.0018 | 0.9977 |
| 2.766 | 8.35 | ÷ | 8.41 | 0.84 | ÷ | 0.85 | 2.65 | ÷ | 2.69 | 0.0021 | 0.9978 |
| 3.076 | 11.33 | ÷ | 11.43 | 0.89 | ÷ | 0.91 | 2.53 | ÷ | 2.59 | 0.0029 | 0.9982 |
| 3.084 | 10.86 | ÷ | 10.95 | 0.87 | ÷ | 0.89 | 3.11 | ÷ | 3.18 | 0.0022 | 0.9982 |
| 4.074 | 15.88 | ÷ | 16.02 | 1.19 | ÷ | 1.21 | 2.89 | ÷ | 2.94 | 0.0048 | 0.9979 |
| 4.087 | 15.07 | ÷ | 15.20 | 1.11 | ÷ | 1.13 | 3.75 | ÷ | 3.83 | 0.0045 | 0.9976 |
| 4.132 | 14.71 | ÷ | 14.82 | 1.08 | ÷ | 1.09 | 2.93 | ÷ | 2.98 | 0.0028 | 0.9985 |
| 4.354 | 15.63 | ÷ | 15.77 | 1.14 | ÷ | 1.16 | 3.24 | ÷ | 3.30 | 0.0052 | 0.9979 |
| 4.529 | 17.05 | ÷ | 17.21 | 1.30 | ÷ | 1.32 | 2.88 | ÷ | 2.94 | 0.0055 | 0.9978 |
| 5.852 | 19.26 | ÷ | 19.38 | 1.44 | ÷ | 1.46 | 4.21 | ÷ | 4.25 | 0.0042 | 0.9983 |
| 5.895 | 19.38 | ÷ | 19.54 | 1.37 | ÷ | 1.39 | 3.77 | ÷ | 3.82 | 0.0058 | 0.9981 |
| 6.168 | 18.98 | ÷ | 19.17 | 1.36 | ÷ | 1.39 | 2.87 | ÷ | 2.92 | 0.0091 | 0.9973 |
| 7.076 | 20.64 | ÷ | 20.86 | 1.36 | ÷ | 1.39 | 3.33 | ÷ | 3.39 | 0.0123 | 0.9963 |

| | | | | | | | | | | |
|---|---|---|---|---|---|---|---|---|---|---|
| 7.620 | 20.47 | ÷ | 20.75 | 1.52 | ÷ | 1.55 | 2.33 ÷ 2.39 | | 0.0180 | 0.9951 |
| 7.983 | 21.33 | ÷ | 21.58 | 1.61 | ÷ | 1.64 | 2.92 ÷ 2.98 | | 0.0137 | 0.9965 |
| 8.345 | 21.71 | ÷ | 21.97 | 1.65 | ÷ | 1.68 | 2.81 ÷ 2.86 | | 0.0136 | 0.9964 |

**Table 1. Estimated values of parameters, RMSE, and determination coefficient $r^2$ for ENM with Tang's solution at different injection flow rates for mass transport.**

| Injection flow rate $Q_0$ (m$^3$s$^{-1}$)×10$^{-6}$ | Convective velocity $u_f$ (ms$^{-1}$)×10$^{-3}$ | | | Dispersion $D_f$ (ms$^{-2}$)×10$^{-3}$ | | | Exchange rate coefficient $\alpha$ (s$^{-1}$)×10$^{-3}$ | | | RMSE | $r^2$ |
|---|---|---|---|---|---|---|---|---|---|---|---|
| 1.835 | 2.20 | ÷ | 2.91 | 1.91 | ÷ | 1.95 | 6.27 | ÷ | 6.59 | 0.0065 | 0.9997 |
| 2.325 | 1.74 | ÷ | 2.73 | 1.82 | ÷ | 1.91 | 5.39 | ÷ | 9.26 | 0.0098 | 0.9992 |
| 2.462 | 0.35 | ÷ | 0.52 | 2.42 | ÷ | 2.57 | 2.25 | ÷ | 2.33 | 0.0138 | 0.9984 |
| 2.605 | 0.44 | ÷ | 0.54 | 2.33 | ÷ | 2.40 | 0.74 | ÷ | 0.77 | 0.0073 | 0.9995 |
| 2.680 | 2.18 | ÷ | 2.95 | 1.77 | ÷ | 1.83 | 5.68 | ÷ | 8.31 | 0.0030 | 0.9998 |
| 2.800 | 0.36 | ÷ | 0.79 | 2.53 | ÷ | 2.68 | 3.54 | ÷ | 3.72 | 0.0213 | 0.9982 |
| 2.847 | 1.73 | ÷ | 3.16 | 1.98 | ÷ | 2.06 | 4.95 | ÷ | 13.45 | 0.0283 | 0.9978 |
| 3.003 | 2.34 | ÷ | 2.87 | 2.24 | ÷ | 2.32 | 5.33 | ÷ | 6.55 | 0.0033 | 0.9998 |
| 3.998 | 2.56 | ÷ | 2.75 | 6.63 | ÷ | 6.80 | 2.05 | ÷ | 2.11 | 0.0150 | 0.9993 |
| 4.030 | 2.60 | ÷ | 2.83 | 7.18 | ÷ | 7.36 | 1.42 | ÷ | 1.52 | 0.0147 | 0.9993 |
| 4.217 | 3.85 | ÷ | 4.56 | 8.92 | ÷ | 9.29 | 4.86 | ÷ | 5.77 | 0.0228 | 0.9945 |
| 4.225 | 2.43 | ÷ | 2.64 | 7.53 | ÷ | 7.84 | 1.64 | ÷ | 1.80 | 0.0251 | 0.9987 |
| 4.471 | 2.30 | ÷ | 3.13 | 9.18 | ÷ | 9.50 | 1.06 | ÷ | 1.33 | 0.1115 | 0.9957 |
| 5.837 | 3.51 | ÷ | 4.13 | 4.95 | ÷ | 5.36 | 0.61 | ÷ | 0.79 | 0.2360 | 0.9872 |
| 5.880 | 2.71 | ÷ | 3.10 | 4.23 | ÷ | 4.60 | 0.04 | ÷ | 0.05 | 0.1997 | 0.9926 |
| 6.445 | 4.71 | ÷ | 5.12 | 6.18 | ÷ | 6.81 | 1.49 | ÷ | 1.54 | 0.2156 | 0.9863 |
| 7.056 | 8.15 | ÷ | 8.46 | 10.05 | ÷ | 10.74 | 5.63 | ÷ | 6.00 | 0.0694 | 0.9951 |
| 7.959 | 9.64 | ÷ | 10.11 | 18.40 | ÷ | 19.47 | 10.92 | ÷ | 11.55 | 0.0662 | 0.9971 |
| 8.971 | 13.40 | ÷ | 13.79 | 24.57 | ÷ | 25.82 | 15.35 | ÷ | 15.85 | 0.0303 | 0.9985 |
| 12.364 | 11.01 | ÷ | 11.67 | 21.97 | ÷ | 22.63 | 5.23 | ÷ | 5.25 | 0.0631 | 0.9939 |
| 12.595 | 13.71 | ÷ | 14.26 | 26.65 | ÷ | 27.61 | 9.26 | ÷ | 9.41 | 0.0426 | 0.9955 |

**Table 2. Estimated values of parameters, RMSE, and determination coefficient $r^2$ for ENM with Tang's solution at different injection flow rates for heat transport.**

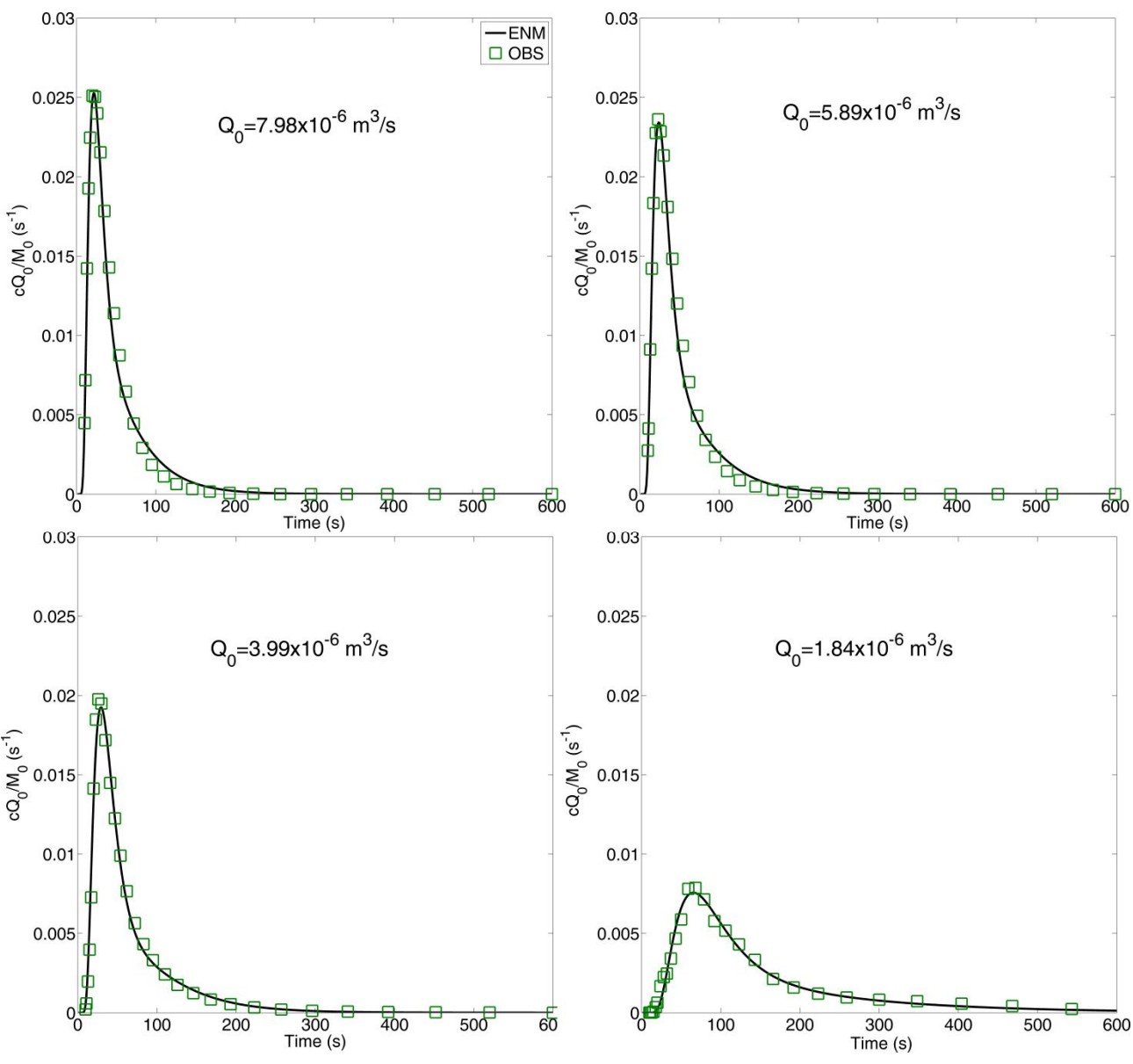


**Figure 4. Fitting of BTCs at different injection flow rates using ENM with Tang's solution for mass transport. Green square curve is the observed specific mass flux at the outlet port, continuous black line is the simulated specific mass flux.**

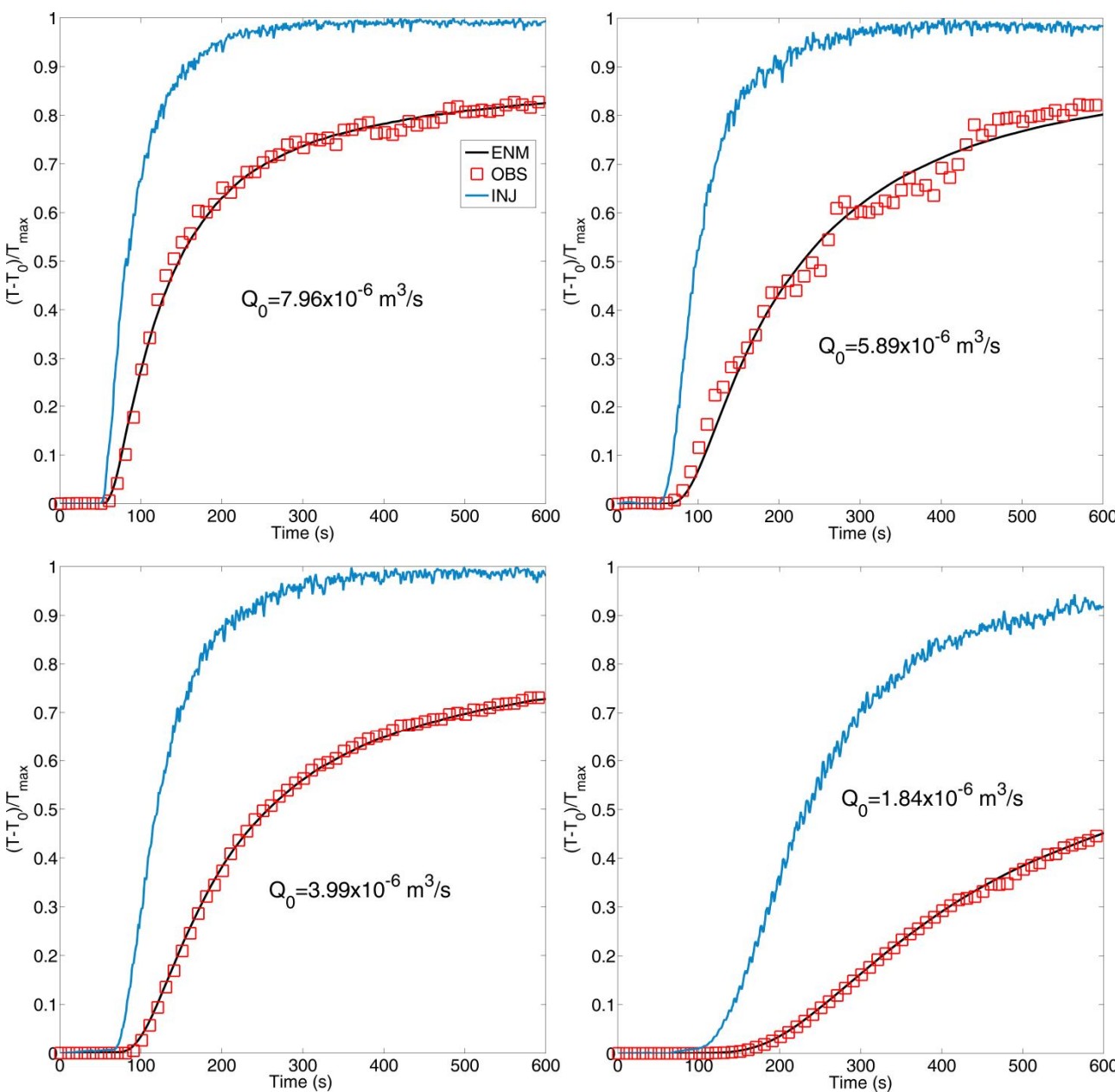


**Figure 5. Fitting of BTCs at different injection flow rates using ENM with Tang's solution for heat transport. The blue curve is the temperature observed at the inlet port used as the temperature injection function, the red square curve is the observed temperature at the outlet port, the black continuous curve is the simulated temperature at the outlet port.**

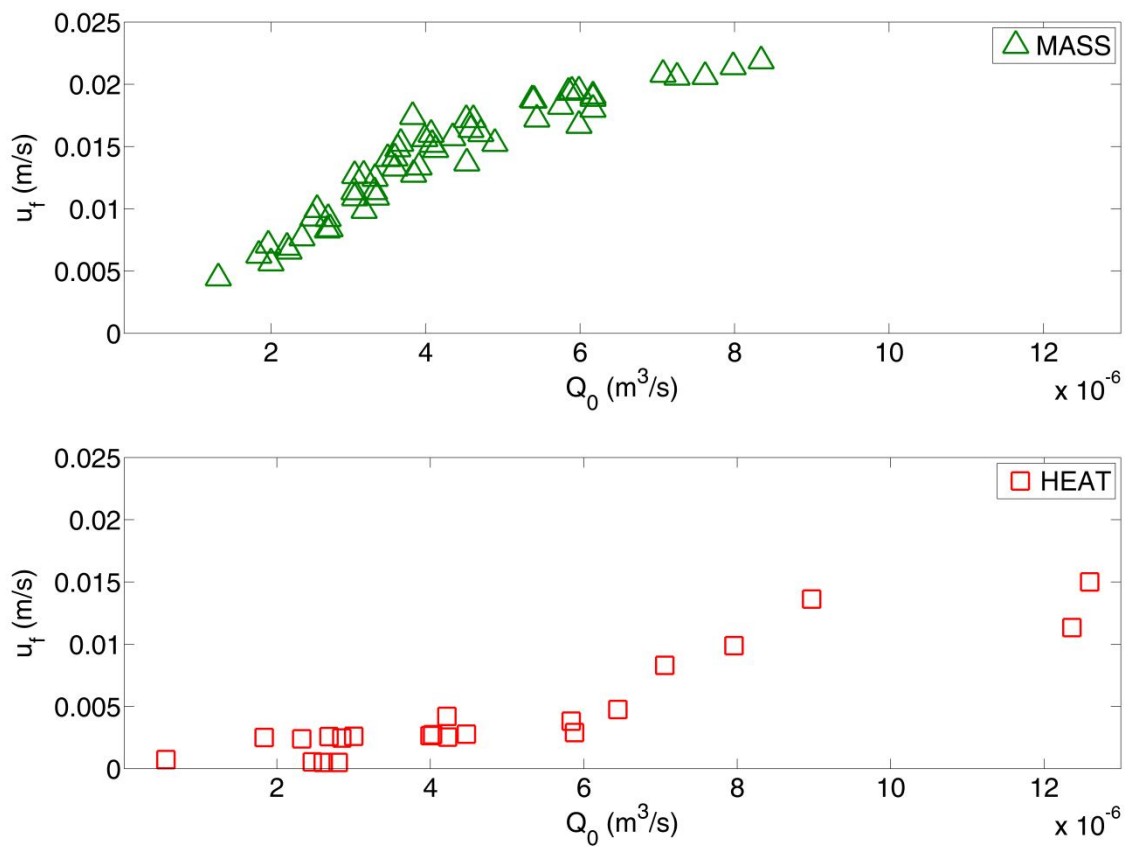


**Figure 6. Velocity $u_f$ (m·s$^{-1}$) as function of the injection flow rate $Q_0$ (m$^3$s$^{-1}$) for ENM with Tang's solution for both mass and**
**heat transport.**

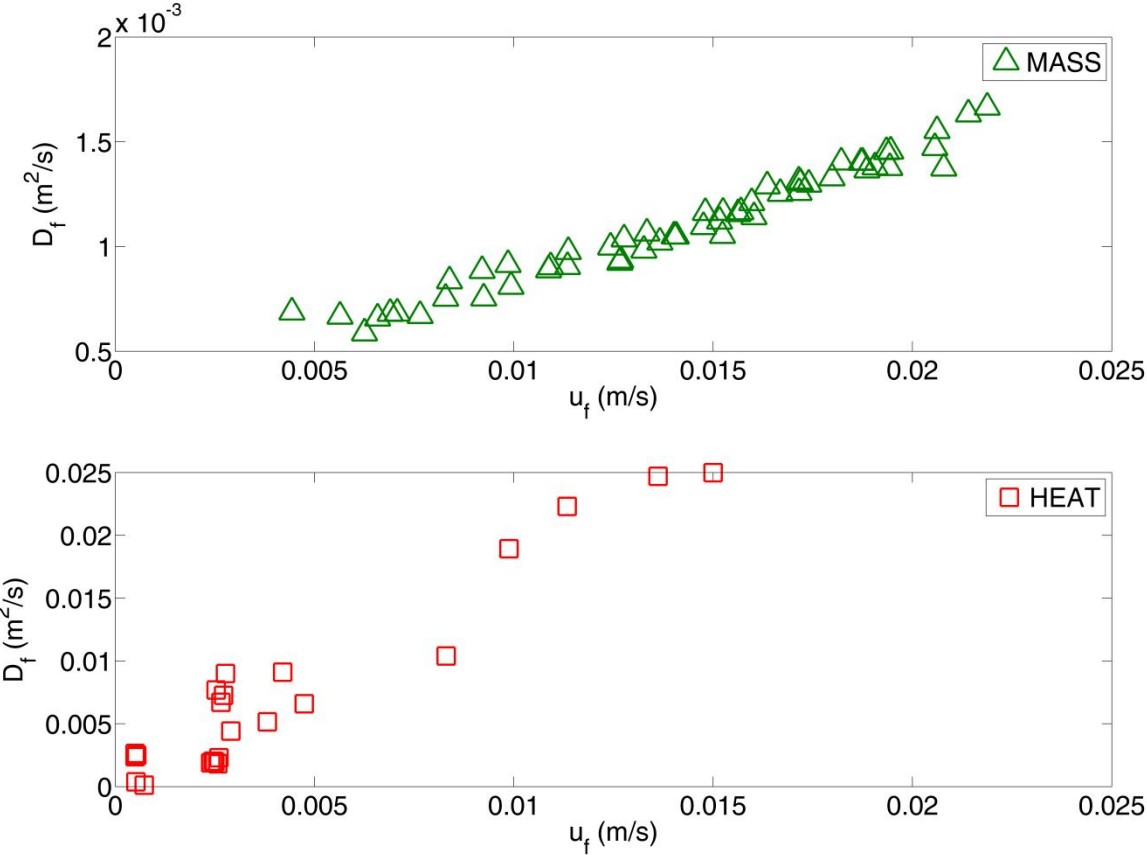


**Figure 7. Dispersion $D_f$ (m·s$^{-2}$) as function of velocity $u_f$ (m·s$^{-1}$) for ENM with Tang's solution for both mass and heat transport.**


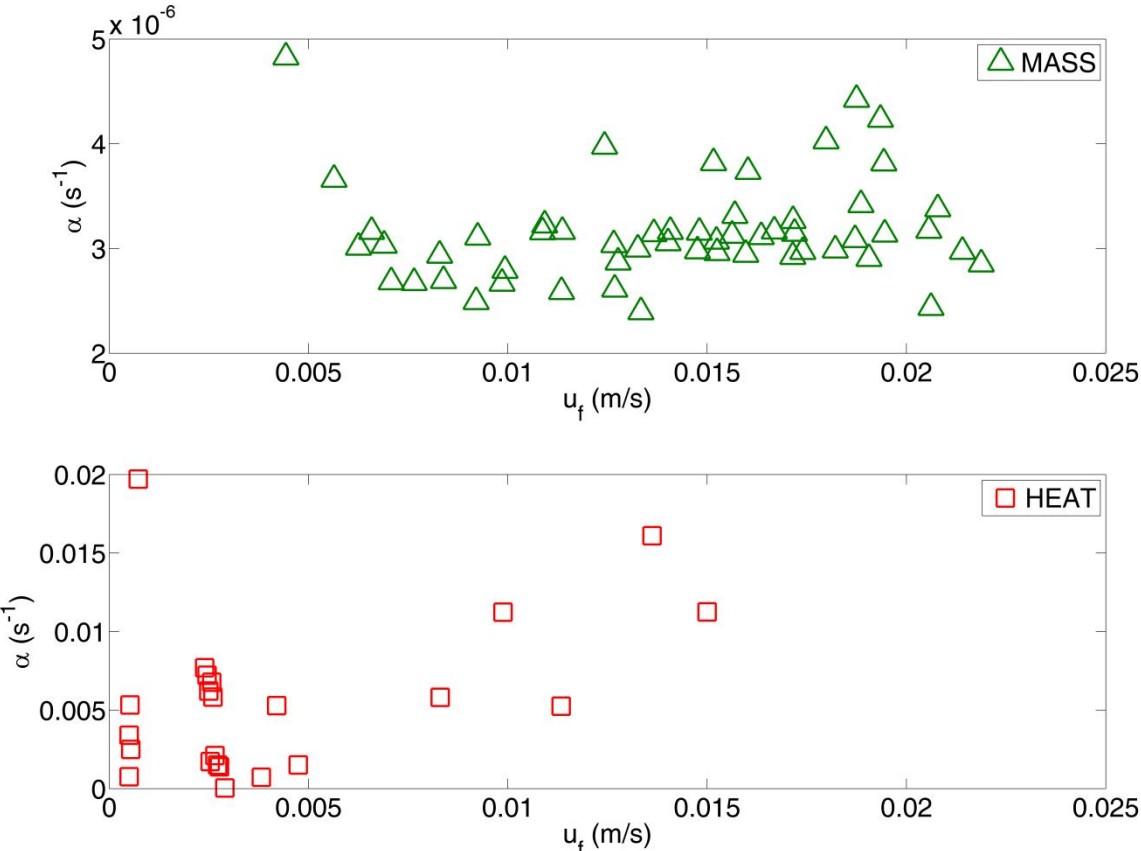


**Figure 8. Transfer coefficient α (s⁻¹) as function of velocity $u_f$ (m·s⁻¹) for both mass and heat transport.**

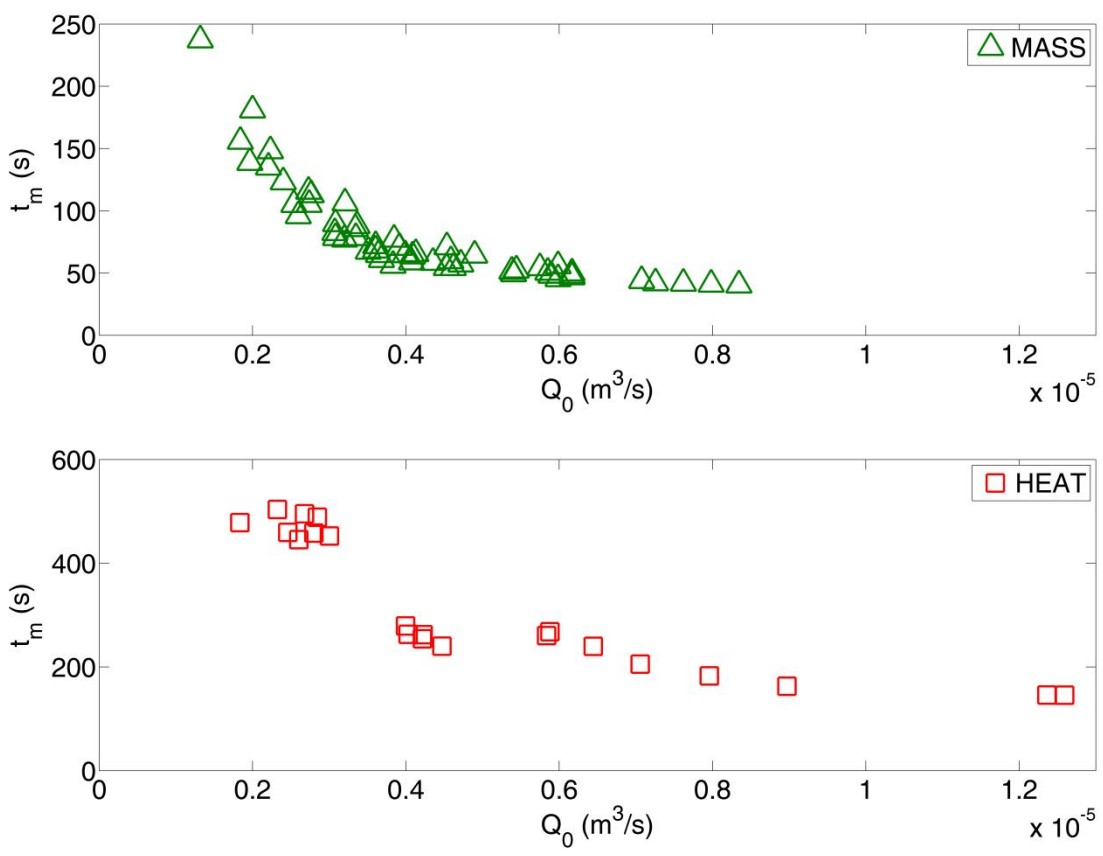


**Figure 9. Mean travel time $t_m$ (s) as function of injection flow rate for both mass and heat transport.**


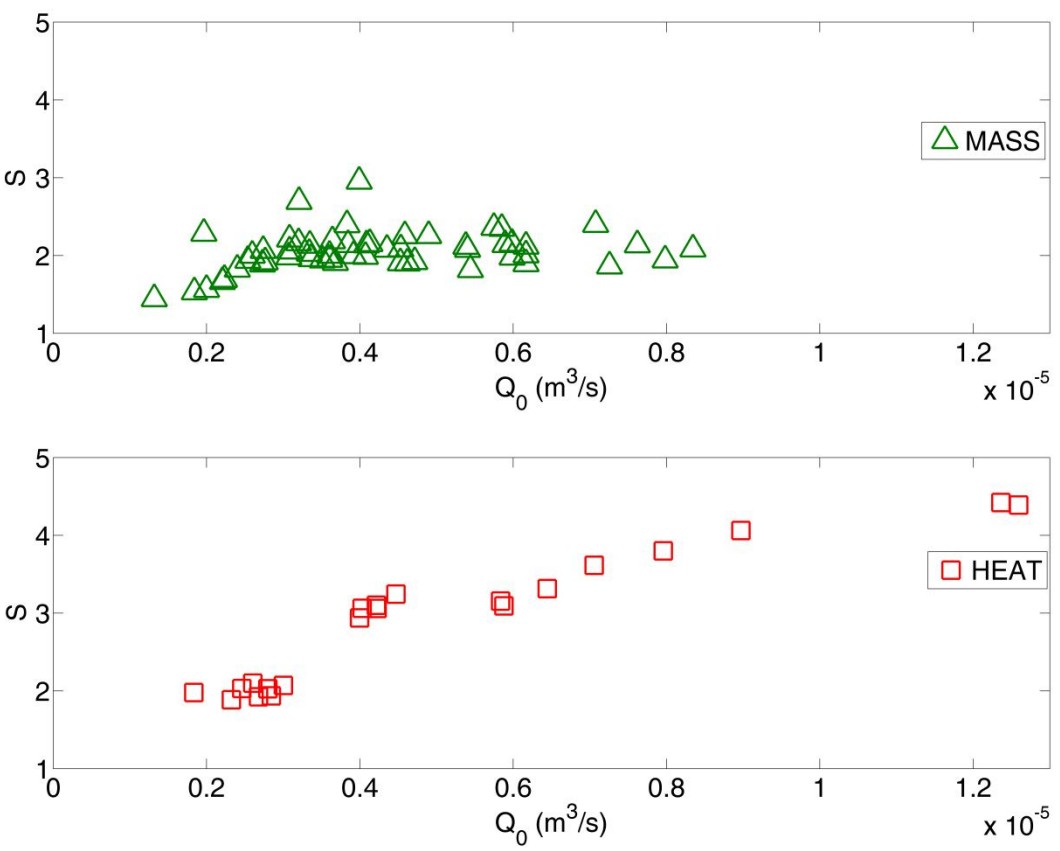


**Figure 10. Skewness as function of injection flow rate for both mass and heat transport.**

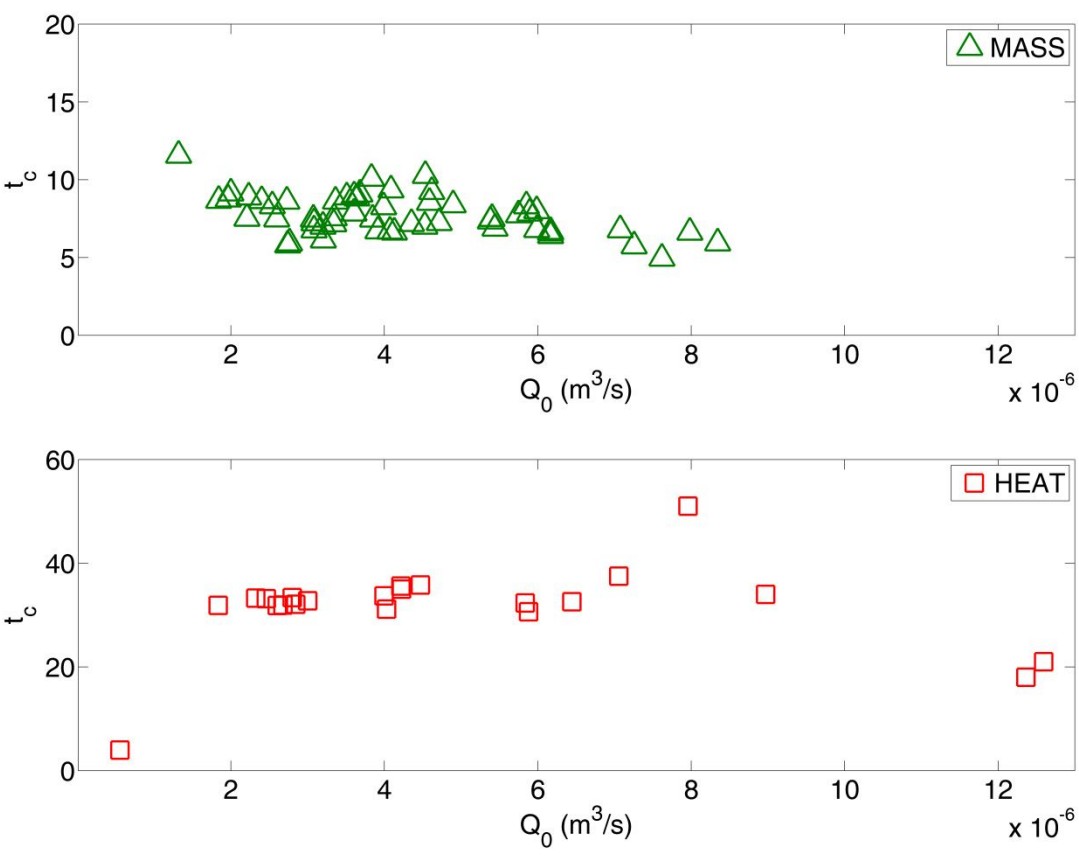


**Figure 11. Tailing character $t_c$ as function of injection flow rate for both mass and heat transport.**



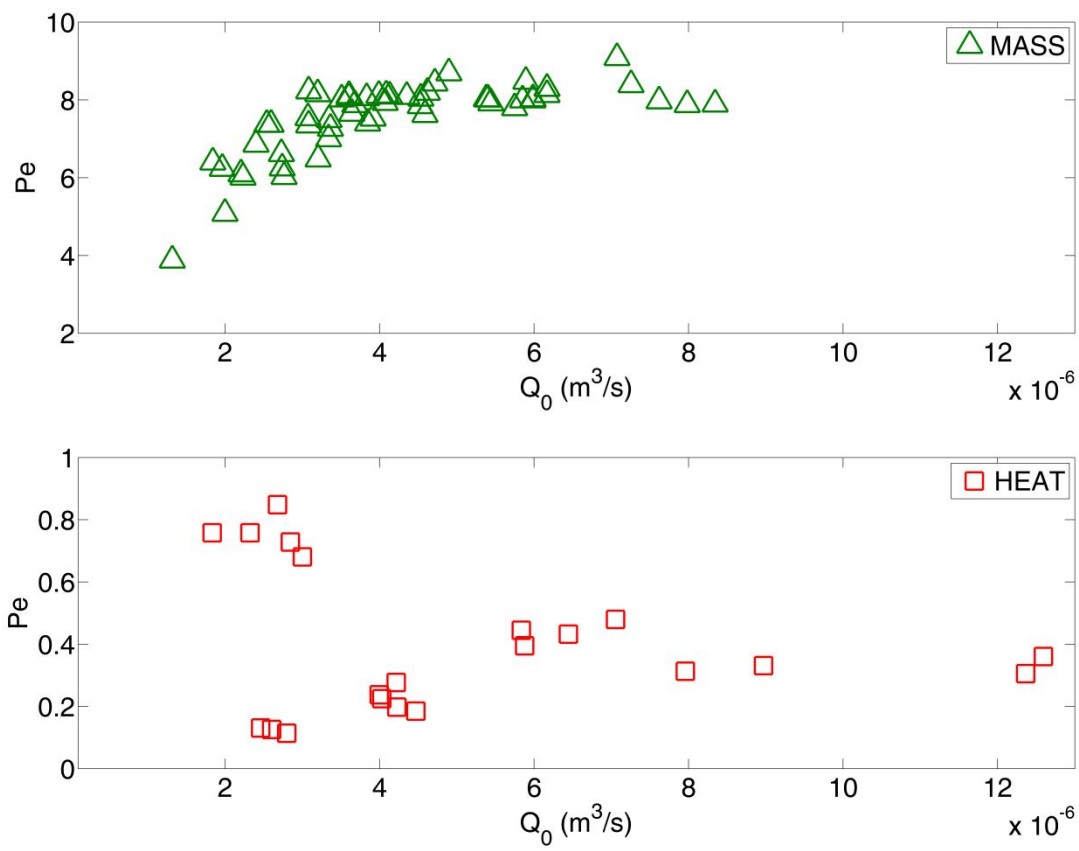


**Figure 12. Peclet number as function of injection flow rate $Q_0$ (m$^3$s$^{-1}$) for both mass and heat transport.**

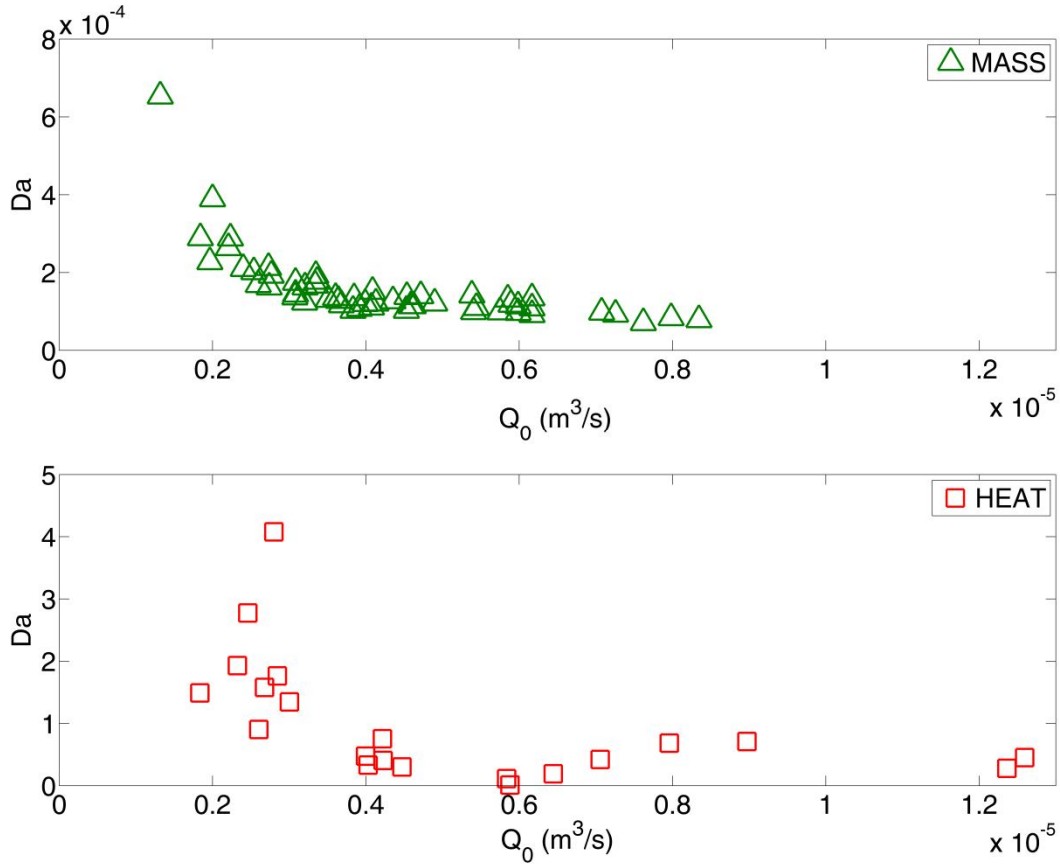


**Figure 13. Da number as function of injection flow rate $Q_0$ ($m^3s^{-1}$) for both mass and heat transport.**

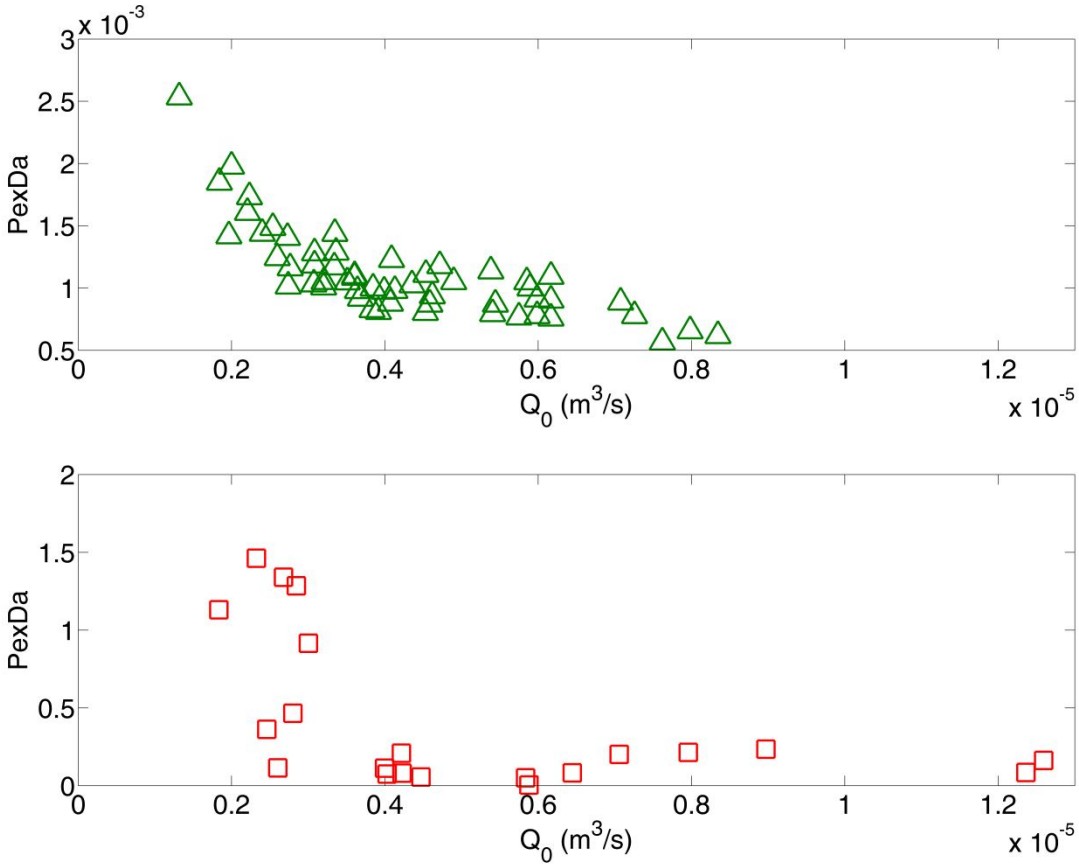


**Figure 14. Pe×Da number as function of injection flow rate $Q_0$ (m³s⁻¹) for both mass and heat transport.**

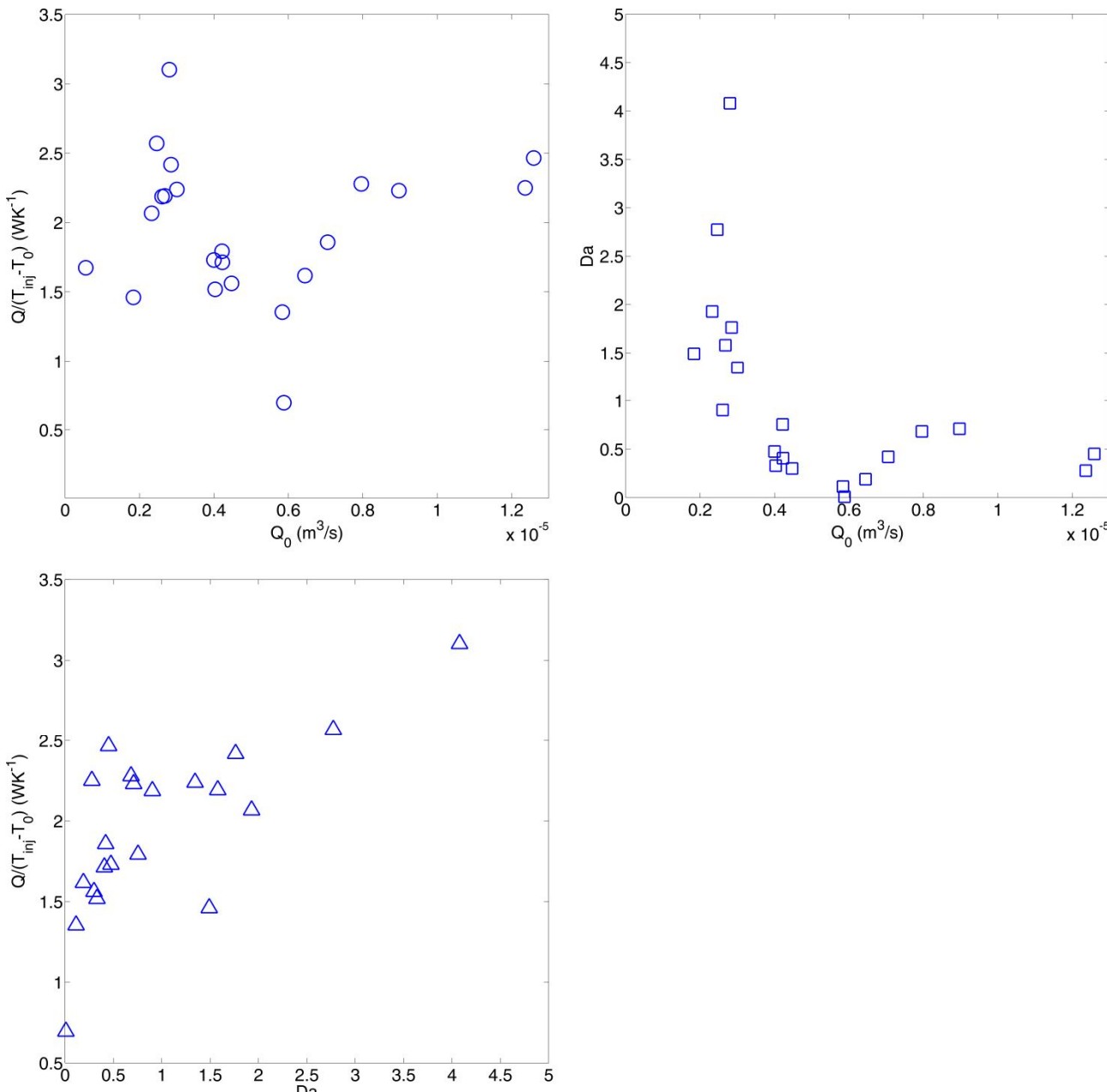


Figure 15. Heat power exchanged per difference temperature unit $\dot{Q}/(T_{inj}-T_0)$ as function of injection flow rate $Q_0$ (m³s⁻¹) (a),
Damköhler number *Da* as function of injection flow rate (b), power exchanged per difference temperature unit as function of
Damköhler number (c).





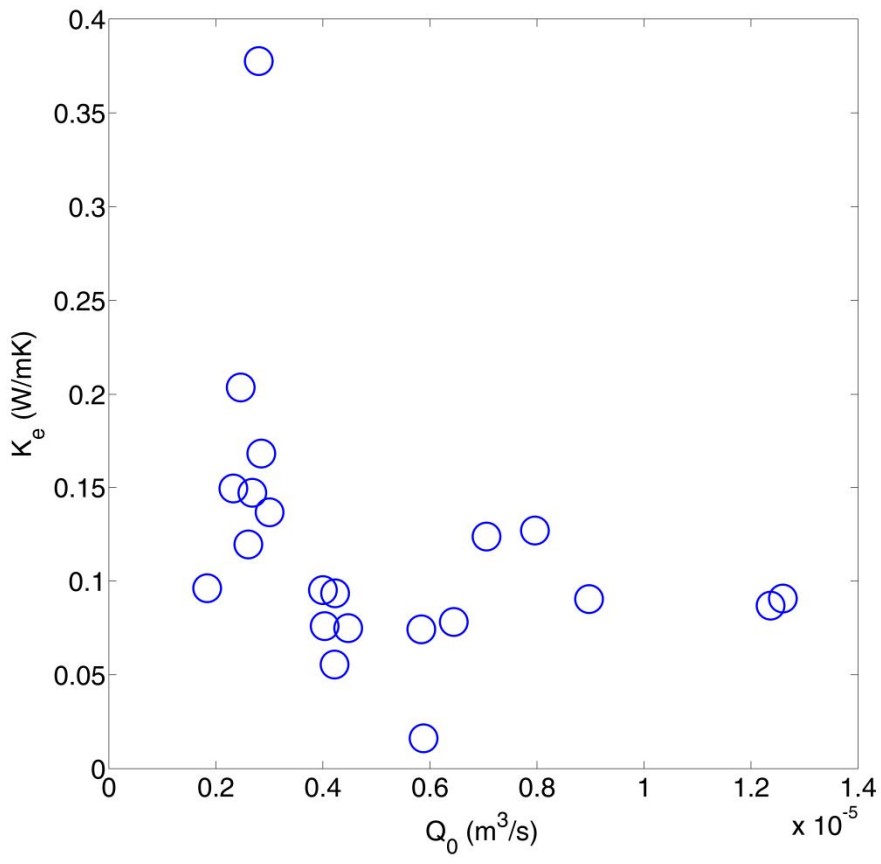

**Figure 16. Effective thermal conductivity $k_e$ (Wm$^{-1}$K$^{-1}$) as function of injection flow rate $Q_0$ (m$^3$s$^{-1}$).**

