# Peer review of "LABORATORY EXPERIMENTAL INVESTIGATION OF HEAT TRANSPORT IN"

_Nonlinear Processes in Geophysics, 2016_

## Referee Comment (RC1) · Anonymous Referee #1 · 6 Nov 2016

The paper aims to investigate experimentally heat transport in fractured rock networks. The results are interpreted through comparing the Explicit Network Model (ENM).

This paper seeks to fill some of the gap in the literature as there are very few such studies to date. The paper begins with an extensive survey of the current state of the art in this field. The subject is important because it pertains to an important source of renewable energy, namely geothermal energy transported particularly by ground water.

A non-linear flow model with a Forchheimer type correction term is used describe the fluxes of mass and heat flow in a single fracture-matrix setup, yielding transport PDE's for concentration and heat (temperature). Three characteristic time scales are identified, yielding non-dimensional system parameters – Pe (Peclet number), Da (Damkohler number).

[Figure]

The ENM is based on an analogy with electrical circuits; here the fluxes are analogous to electrical current. Then the 1st and 2nd Kirchhoff's laws are applicable. This leads to breakthrough curves (BTC) that describe the concentration c(t) and the heat (tempertaure) T(t) in the fracture as a function of time.

Experiments are conducted in an already known setup. The parameters that control the mass and heat transport were estimated using the ENM. The ENM model match the results in most cases fairly well.

The main conclusions are that rock-fracture size ratio plays an important role in the fluid to solid heat transfer processes. It is also concluded that it is not efficient to store thermal energy in rocks with high fracture density. Other points discussed pertain to optimal conditions for thermal exchange in a fractured network, and the non-Fickian behavior of solute BTC's.

Overall, the paper tells a good story. The modeling and experiments is well described and conclusions drawn are reasonable.

In the conclusions, the authors should say a few words about future directions. What about more complex fracture networks? What are the range of pore sizes and porosity and permeability that the ENM can be applied to?

Minor point: the quality of the figures should be improved.

---

## Referee Comment (RC2) · Anonymous Referee #2 · 11 Nov 2016

**Review of manuscript:**

**"LABORATORY EXPERIMENTAL INVESTIGATION OF HEAT TRANSPORT IN FRACTURED MEDIA"**
**By Claudia Cherubini et al.**

Submitted to Non Linear Processes in Geophysics.
Author(s): Claudia Cherubini, Nicola Pastore, Concetta I. Giasi and Nicoletta Maria Allegretti.
MS No.: NPG-2016-54
doi: 10.5194/npg-2016-54

**General comments**

This paper presents an actual subject (heat transport in fractured media) with economical and ecological implications (geothermal energy plants as one of the renewable energies). The authors present an experiment which is very interesting and original due to their experimental apparatus and setup. The theory of the subject is clearly stated by the authors with a detailed and wide description.

The main objective of the paper is to study the mass and heat fluxes in a fractured media which behaviour is not well understood and it is an important topic for geothermal energy extraction. The authors use the explicit network model (ENM) to analyze the behaviour by means of several parameters shown in different figures (concentration and heat as a function of time, evolution of convective velocity, the dispersion coefficient and several time scales and non-dimensional numbers).

The authors of this work establish the basis to analyze the optimal conditions for thermal exchange in fractured media; they also deduce that the thermal dispersion dominates heat transport dynamics and the role of the rock-fracture size ratio.

The authors should explain better the applications of their conclusions at sections Results and Discussion and Conclusions. Explain how their results are important for industrial situations (energy plants) and if their conclusions establish any limitation to the use or to the characteristics of geothermal energy plants.

This paper should undergo a minor revision and some technical corrections before being considered for publication.

**Specific comments**

**Abstract**

- **Page 1, Lines 14-15.** You say: "One of the major limitations related to the choice of installing low enthalpy geothermal power plants regards the initial investment costs." Is it possible you add and describe more problems and limitations?

**Introduction**

- **Page 5, Lines 123-138.** You mention several dilemmaes. How does your study help to clarify these problems?
- **Page 5, Line 143.** What is the tortuosity factor?

**Section "Theoretical background".**
**Subsection "Nonlinear flow"**

- **Page 7, Line 187.** What is the hydraulic head?

**Section "Theoretical background".**
**Subsection "Heat transfer by water flow in single fractures"**
- **Page 8, Line 221.** Explain better the meaning of $D_e$ and $k_e$.
- **Page 9, Line 243**. What is the function $\theta_m$?
- **Page 10, Line 259.** Explain what is the residence time.
- **Page 11, Lines 288.** The Peclet number you define, is it not the ratio between dispersive ($t_d$) to convective ($t_u$) transport times?

**Section "Theoretical background".**
**Subsection "Explicit network model"**

- **Page 12, Lines 311.** Introduce or describe the main characteristics of the ENM model.
- **Page 13, Lines 329 and 330.** Which is the subscript of the summation in equation (35) and line 330? The same applies to equation (38) and line 354 in page 14.

**Section "Material and methods"**
**Subsection "Flow experiments"**

- **Page 15, Line 381.** Is the average hydraulic head the same that head loss?

**Section "Material and methods"**
**Subsection "Solute and temperature tracer tests"**

- **Page 16, Line 394-395.** Describe what is the instantaneous source assumption and why you can use it.

**Section "Results and discussion"**
**Subsection "Flow characteristics"**

- **Page 16, Line 407.** Why have the linear and nonlinear terms been assumed equal?
- **Page 16. Line 410.** Explain better how to get the equation (42). Describe also what is the meaning of $Q_0$ and $R_i$ with $i=1-9$ in a new line.
- **Page 16. Line 412.** Explain better how to get the equation (43).
- **Page 17. Line 417.** Which is the meaning and importance of the critical flow rate, $Q_{crit}$?
- **Page 17. Lines 417-418.** Could you explain better why the critical flow rate, $Q_{crit}$ can be determined in correspondence of $Fo=1$ as the ratio between $a$ and $b$?

**Section "Results and discussion"**
**Subsection "*Fitting of breakthrough curves and interpretation of estimated model parameters*"**

- **Page 17. Lines 425 to 428**. Is there any adimensional number to do easier the comparison of these different experiments.
- **Page 17. Lines 430-431**. Why are the transport parameters $u_f$, $D_f$ and $\alpha$ assumed equal for all branches?
- **Page 18**. **Line 445.** Why the characteristic length is equal to 0.601?
- **Page 19. Line 493**. Is the mean travel time the same magnitude that the mean residence time (Y-axis of Figure 8).
- **Page 21. Lines 543 to 547**. Which could be the practical use of the conclusions described in these lines?
- **Page 22. Line 558-559**. Could you explain how the gradient of Tm is evaluated according to Equation (16)?
- **Page 22. Lines 566 and 567**. Which are the implications of your conclusion that there is a solid thermal resistance which depends on the rock − fracture size ratio?

**Section "Conclusions"**
- **Page 23. Lines 602 and 603.** Could you describe in more detail the optimal conditions for thermal exchange in a fracture network and your future research**?**

**Technical comments**

- There is not section numbering. Number all sections and subsections as follows:
  **1 Introduction**
  **2 ...............**
      **2.1 .............**
      **2.2 ……**
  **3 ...............**

- *Be careful with the use of subscripts. For a given magnitude, sometimes you use subscripts and other not (for example, Da). Revise the whole text.*

- You would write equations from (39) to (53) in a larger size.

**Abstract**

- **Page 1, Lines 13.** Add or: "cooling of industrial processes, food drying systems *or* desalination".

**Introduction**

- **Page 6.** Add a new paragraph at the end of the Introduction section to summarize your paper as follows: *"In section 1 we shows.....Section 2 describe....".*

**Section "Theoretical background".**
**Subsection "Nonlinear flow"**

- **Page 7, Line 186.** Add the meaning of the coefficients $\mu$, $u_f$, $k$, $\rho$ and $p$.
- **Page 7, Line 189.** You write: "The coefficients $a$ (TL-1) and $b$ (TL-2) represent…." I think that is not $a$ and $b$ but $a'$ y $b'$.

**Section "Theoretical background".**
**Subsection "Heat transfer by water flow in single fractures"**
- **Page 8, Line 217.** In equation (8) you write $C_m$ in capital letter but in equation (6) you write $c_m$ in lower case letter. Are they the same coefficients, $C_m$ and $c_m$, or are different? If they are the same, use the same notation (in capital letter or in lower case letter always). The same applies to the equations (10) and (12). The same applies to lines 247 and 251 of page 9 or in equation (14).
- **Page 9, Line 231**. If you define $u_f$ before, in line 186 of page 7, you must eliminate it in line 231 of page 9.
- **Page 9, Line 244**. You write: " (-) the matrix porosity". What is the symbol (-)?
- **Page 9, Line 248.** You write: "function of time in Laplace space.". Eliminate the point and write something like "as follows".
- **Page 9, Line 250.** After this line, write another one to define the magnitudes $s$, $v$, $L$, $\beta$, $A$ and $B$.
- **Page 9, Line 251 and Page 10, Line 252.** Write these two lines later and after equation (19).
- **Page 10, Lines 253 to 257.** Write these lines before and after equation (14) in line 250.
- **Page 10, Lines 255.** Equation (17) is:

$$A = \frac{\delta}{\sqrt{\theta D_e}}; \theta = \theta_m$$

Why not write

$$A = \frac{\delta}{\sqrt{\theta D_e}}; \theta = \theta_m$$

- **Page 10, Lines 258.** You write: "Furthermore on the basis of these analytical solutions the probability density function of the solute residence time (*PDF*)". Change this line as follows:

"Furthermore on the basis of these analytical solutions the probability density function (PDF) of the solute residence time."

- **Page 11, Lines 271 and 274.** In equation (26) you write $C_w$ in capital letter but in equation (23) you write $c_w$ in lower case letter. Are they the same coefficients, $C_w$ and $c_w$, or are different? If they are the same, use the same notation (in capital letter or in lower case letter always).
- **Page 11, Lines 278.** Eliminate the semicolon at the end of equation (25).
- **Page 11, Lines 279.** Rewrite equation (26):

$$A = \frac{\delta}{\sqrt{\theta D_e}}; \quad \theta = \frac{\rho_m C_m}{\rho_f C_f}, \quad D_e = \frac{k_e}{\rho_w C_w}$$

For example: $A = \ldots\ldots$ where $\theta = \ldots.$ and $D_e = \ldots.$

- **Page 12, Lines 294 to 298.** First, define Damköhler number and use equation (32). Second, write your line "Note that the inverse of $t_e\ldots\ldots$" and define $\alpha$ with equation (31). Again, be careful with subscript: you write $Da$ in equation (32) and $D_a$ in line 296, 299, 302, 303…..In equation (33), no subscripts, etc.

**Section "Theoretical background".**
**Subsection "Explicit network model"**

- **Page 12, Line 312 and Page 13, Lines 316 and 317-318.** Use another notation to write $SF\ j$ (this is a bit confusing). Something like j-th simple fracture or j-th SF.
- **Page 12, Line 312.** You write "schematized by a 1d – pipe element". Use better 1 D-pipe element.
- **Page 13, Lines 319 and 322.** Write 2D instead of 2d.
- **Page 14, Line 339.** Write the convolution operator without parenthesis: *.
- **Page 14, Line 340.** The notation for the inverse Laplace transform operator, $L^{-1}$, could be mistaken for the characteristic length, L. Use another notation.
- **Page 14, Lines 348-349.** Modify the phrase as follows: Where $T_0$ (K) is the initial temperature, $T_{inj}$ (K) is the temperature injection function and $P_{H,j}$ is the heat distribution probability.

**Section "Material and methods"**
**Subsection "Description of the experimental apparatus"**

- **Page 15, Lines 375.** Verify the writing of the units: $\mu S\ cm^{-1}$.

**Section "Results and discussion"**
**Subsection "Flow characteristics"**

- **Page 16, Line 406.** Write 2D instead of 2d.
- **Page 17. Line 417.** Write $Fo=1$ in italics.

**Section "Results and discussion"**
**Subsection "*Fitting of breakthrough curves and interpretation of estimated model parameters*"**

- **Page 17. Line 430.** It is better to indicate the number of the section you refer, and not "the previous section".
- **Page 17. Line 433**. Review the number of the equations quoted: Are not equations 36 and 37?
- **Page 17**. **Lines 441 to 444**. You would quote Tables 1 and 2 in this paragraph to verify the conclusions deduced in these lines.
- **Page 18**. **Lines 448 to 451**. The same as before: you would quote Tables 1 and 2 in this paragraph to verify the conclusions deduced in these lines.
- **Page 18. Line 468.** You write: "whereas $u_f$ is influenced". Add a blank space.

**References**

Review all the bibliographic references.
Review the NPG format for references because there are several references that are not well written (the punctuation, the initial page or the final page, the position of the year is not at the end or the review is not written in italics). For example, the following references are not correct:

- Auradou, H., Deazerm G., Boschan, A., Hulin J., Koplik, J., 2006. Flow channeling in a single fracture induced by shear displacement. Geothermics 35 575-588.
- Becker, M.W. and Shapiro, A. M., 2003. Interpreting tracer breakthrough tailing from different forcedgradient tracer experiment configurations in fractured bedrock. Water Resources Research. 39(1):1024. pages.
- Cherubini, C., Pastore, N., 2011. Critical stress scenarios for a coastal aquifer in southeastern Italy. Natural Hazards and Earth System Science. 11 (5) p. 1381-1393.

And more.

I do not understand the following reference:

- Geiger, S. and Emmanuel, S., 2010. Non-fourier thermal transport in fractured geological media. Water Resources Research, 46, xvii, 26, 27, 168.

The following references are not quoted in the text but they are in the list:

- De Hoog, F.R., Knight, J.H., Stokes, A.N., 1982. An improved method for numerical inversion of Laplace transforms. SIAM J. Sci. Stat. Comput. 3 (3), 357-366.
- Hasler, A., Gruber, S., Font, M., Dubois, A., 2011. Advective heat transport in frozen rock cleft – Conceptual model, laboratory experiments and numerical simulation…..pages?

You quote in the text some references but there are no corresponding bibliographic references in the list:

- Neuville et al. (2010)

- All quotes in page 4 are not in the reference list.

- Hopmans et al. (2002).
- Sauty et al. (1982)
- Papadopulos and Larson, 1978.
- Smith and Chapman, 1983.
- Molson et al., 1992.

- De Marsily, 1986.
- Anderson, 2005.
- Hatch et al., 2006.
- Keery et al., 2007.
- Vandenbohede et al., 2009.
- Vandenbohede and Lebbe, 2010.
- Rau et al., 2010.
- Green et al., 1964.
- Wu et al., 2010.

You mention the following reference (Cherubini et al., 2013) in the text (page 9, line 233 and page 18, line 457) but you write four different references in the list: 2013a, 2013b, 2013c and 2013d. Which is the reference you want to quote? All?

Use the same line spacing in the reference list (see page 25 from line 657 to 668 and page 26 from line 669 to 674).

The following reference (page 26, line 673 to 674):

Martinez, A. R., Roubinet, D., Tartakovsky, D. M., 2014. Analytical models of heat conduction in fractured rocks. J. Geophys. Res. Solid Earth, 119.

is not in a separate line from the previous bibliographic reference.

**Tables**
- **Table 1 and 2**
  Modify the caption to include the definition of $Q_o$, $u_f$, $D_f$ and $\alpha$.

**Figures**
- **Figure 1.** It would be interesting to add a photograph of the experimental setup in Figure 1.
- **Figure 2.** Modify the caption to include the definition of $Q_o$, $Q_1$ and $Q_2$. Is it possible to use a frontal figure and not this tilted one? May be, it is also possible to say that is an extension of the rock matrix in Figure 1.
- **Figure 3.** The label of the Y-axis in the upper left figure is different from the others Y-labels.
- **Figure 4**. Describe the meaning of the blue curve in the caption.
- *Figure 7 is not quoted in the text.*
- **Figure 12.** Review the writing of the symbols $D_a$ and $Q_o$ with and without subscripts in the text, in the caption of this figure and in the labels of the axis.
- **Figure 15.** Use smaller symbols as in the previous figures.

---

## Referee Comment (RC3) · Anonymous Referee #3 · 11 Nov 2016

This paper introduces a theoretical analysis of a comple problem involving non Fourier and non Darcy behavior. A high quality experime,tal set up is used for a chosen configuration and results agree with a numerical explicit network model. The physical analysis based on relevant literature is especially well conducted and this pioneering paper will be of interest in the field of geothermal energy production.

I recommend for publication after verifying tying details.

---

## Author Comment (AC1) · 6 Dec 2016

**General comments**

*'The authors should explain better the applications of their conclusions at sections Results and Discussion and Conclusions. Explain how their results are important for industrial situations (energy plants) and if their conclusions establish any limitation to the use or to the characteristics of geothermal energy plants.'*

The Conclusion has been expanded and it has been explained how the results are important for geothermal power development and the limitations/optimal conditions for it have been better addressed.

*'This paper should undergo a minor revision and some technical corrections before being considered for publication.'*

The revisions and technical corrections have been done as explained in the following paragraphs.

**Specific comments**

**Abstract**
**Page 1, Lines 14-15.** *You say: "One of the major limitations related to the choice of installing low enthalpy geothermal power plants regards the initial investment costs." Is it possible you add and describe more problems and limitations?*
This has been done. The following paragraph has been added:

'Geothermal power development is a long, risky and expensive process. It basically consists of successive development stages aimed at locating the resources (exploration), confirming the power generating capacity of the reservoir (confirmation) and building the power plant and associated structures (site development). Different factors intervene in influencing the length, difficulty and materials required for these phases thereby affecting their cost.
One of the major limitations related to the installation of low enthalpy geothermal power plants regards the initial development steps which are risky and the upfront capital costs that are huge.
Most of the total cost of geothermal power is related to the reimbursement of invested capital and associated returns.'

**Introduction**
**Page 5, Lines 123-138.** *You mention several dilemmaes. How does your study help to clarify these problems?*
The following paragraph has been added:
'The present study is aimed at providing a better understanding of heat transfer mechanisms in fractured rocks. Laboratory experiments on mass and heat transport in a fractured rock sample have been carried out in order to analyze the contribution of thermal dispersion in heat propagation processes, the influence of nonlinear flow dynamics on the enhancement of thermal matrix diffusion and finally the optimal conditions for thermal exchange in a fractured network.'

**Page 5, Line 143.** *What is the tortuosity factor?*
The previous version of the paper (as already stated by the referee in the previous comment) did not provide any reference on how the study can help clarify all the conflicting theories concerning thermal

dispersivity. Moreover, there was no direct link to the work done by other scientists as far as heat transfer in fractured formations and the current study since there was the abrupt introduction to all previous studies done by the authors concerning flow and transport in the fractured formation:' In previous studies by Cherubini et al. (2012, 2013a, 2013b, 2013c and 2014) the presence of nonlinear flow and non – Fickian transport in a fractured rock formation has been detected.'

And then the text went on with a long description of all the previous results which can distract the reader from the main focus. This part has been synthetized and it has been more focussed on those results pertinent to the study of heat transfer.

**Section "Theoretical background".**
**Subsection "Nonlinear flow"**
☐**Page 7, Line 187.** *What is the hydraulic head?*
The text has been changed with 'It is possible to express Forchheimer law in terms of hydraulic head $h$ (L)'

**Section "Theoretical background".**
**Subsection "Heat transfer by water flow in single fractures"**
☐**Page 8, Line 221.** Explain better the meaning of $D_e$ and $k_e$.
The following text has been added : The effective diffusion coefficient takes into account the fact that diffusion can only take place through pore and fracture openings because mineral grains block many of the possible pathways. The effective thermal conductivity of a formation consisting of multiple components depends on the geometrical configuration of the components as well as on the thermal conductivity of each.

☐**Page 9, Line 243**. *What is the function $\vartheta_m$?*
$\theta_m$ is not a function but it is the matrix porosity.
The sentence 'Where Da is the apparent diffusion coefficient of the solute in the matrix expressed as function of $\theta_m$' means that Da depends on $\theta_m$ (is a function of)

☐**Page 10, Line 259.** *Explain what is the residence time.*
This sentence has been added: 'Defined the residence time as the average amount of time that the solute spends in the system'

☐**Page 11, Lines 288.** The Peclet number you define, is it not the ratio between dispersive ($t_d$) to convective ($t_u$) transport times?
This is right, so the definition of Peclet number has been corrected as 'Peclet number $P_e$ is defined as the ratio between dispersive ($t_d$) and convective ($t_u$) to transport times'

**Section "Theoretical background".**
**Subsection "Explicit network model"**
☐**Page 12, Lines 311.** Introduce or describe the main characteristics of the ENM model.
The following sentence has been added:
'The 2 – D Explicit Network Model (ENM) depicts the fractures as 1 – D pipe elements forming a 2 – D pipe network and therefore expressly takes the fracture network geometry into account. The ENM model permits to understand the physical meaning of flow and transport phenomena and therefore to obtain a more accurate estimation of flow and transport parameters'

**Page 13, Lines 329 and 330.** Which is the subscript of the summation in equation (35) and line 330? The same applies to equation (38) and line 354 in page 14.
The equation has been modified as requested:

$$Q_j = \sum_{i=1}^{n} Q_i \left[ \frac{1}{R_j} \left( \sum_{i=1}^{n} \frac{1}{R_i} \right)^{-1} \right]$$

**Section "Material and methods"**
**Subsection "Flow experiments"**
**Page 15, Line 381.** Is the average hydraulic head the same that head loss?
Yes, so the term 'head loss' has been substituted instead of 'hydraulic head difference' everywhere

**Section "Material and methods"**
**Subsection "Solute and temperature tracer tests"**
**Page 16, Line 394-395.** *Describe what is the instantaneous source assumption and why you can use it.*
The following sentence has been added:
'Due to the very short source release time, the instantaneous source assumption can be adopted which assumes the source of solute as an instantaneous injection (pulse).'

**Section "Results and discussion"**
**Subsection "Flow characteristics"**
**Page 16, Line 407.** *Why have the linear and nonlinear terms been assumed equal?*
The following sentence has been added:

'The resistance to flow of each SF can be evaluated as the square bracket in equation (34). For simplicity the linear and non linear terms have been considered constant and equal for each SF. '

**Page 16. Line 410.** *Explain better how to get the equation (42). Describe also what is the meaning of $Q_0$ and $R_i$ with i=1-9 in a new line.*
The following periods have been added

'The resistance to flow for the whole fracture network $\overline{R}\left(\overline{Q}\right)$ can be evaluated as the sum of the resistance to flow of each *SF* arranged in chain and the total resistance of the parallel branches equal to the reciprocal of the sum of the reciprocal of the resistance to flow of each parallel branch'

'…Where $R_j$ with j = 1 − 9 represents the resistance to flow of each *SF*, $Q_0$ is the injection flow rate, $Q_1$ and $Q_2$ are the flow rates that flowing in the parallel branch 6 and 3 − 4 − 5 respectively.'

**Page 16. Line 412.** *Explain better how to get the equation (43).*
The following sentence has been added:

'The flow rate $Q_1$ is determined in iterative manner using the following iterative equation derived by the equation (35) at the node 3'

**Page 17. Line 417.** Which is the meaning and importance of the critical flow rate, $Q_{crit}$?
The following sentence has been added:

The critical flow rate $Q_{crit}$ which represents the value of flow rate for which $Fo = 1$

☐ **Page 17. Lines 417-418.** *Could you explain better why the critical flow rate, $Q_{crit}$ can be determined in correspondence of Fo=1 as the ratio between a and b?*

The following sentence has been added:
'The linear and nonlinear term are equal respectively to $a = 7.345 \times 10^4$ sm$^{-3}$ and $b=11.65 \times 10^9$ s$^2$m$^{-6}$. Inertial forces dominate viscous ones when the Forchheimer number ($Fo$) is higher than one. $Fo$ can be evaluated as the ratio between the non linear loss $\left(bQ^2\right)$ and the linear loss $\left(aQ\right)$. The critical flow rate $Q_{crit}$ which represents the value of flow rate for which $Fo = 1$ is derived as the ratio between $a$ and $b$ resulting $Q_{crit} = 6.30 \times 10^{-6}$ m$^3$s$^{-1}$.'

**Section "Results and discussion"**
**Subsection "Fitting of breakthrough curves and interpretation of estimated model parameters"**

☐ **Page 17. Lines 425 to 428**. *Is there any adimensional number to do easier the comparison of these different experiments.*
Re number has been used to make the comparison:
'The behavior of mass and heat transport has been compared varying the injection flow rates. In particular 21 tests in the range $1.83 \times 10^{-6}$ - $1.26 \times 10^{-5}$ m$^3$s$^{-1}$ (Re in the range $17.5 - 78.71$) for heat transport have been made and compared with the 55 tests in the range $1.32 \times 10^{-6}$ - $8.34 \times 10^{-6}$ m$^3$s$^{-1}$ (Re in the range $8.2 - 52.1$) for solute transport presented in previous studies.'

☐ **Page 17. Lines 430-431**. *Why are the transport parameters $u_f$, $D_f$ and α assumed equal for all branches?*
'For simplicity the transport parameters $u_f$, $D_f$ and $\alpha$ are assumed equal for all branches of the fracture network.'

☐ **Page 18. Line 445.** *Why the characteristic length is equal to 0.601*?
The following sentence has been added:
'Considering a characteristic length equal to $L = 0.601$ m corresponding to the length of the main path of the fracture network'

☐ **Page 19. Line 493**. *Is the mean travel time the same magnitude that the mean residence time (Y-axis of Figure 8).*
The mean travel time and the mean residence time are synonyms. In order to avoid confusion with two terms, everywhere in the text the term 'residence time' has been used.

☐ **Page 21. Lines 543 to 547**. *Which could be the practical use of the conclusions described in these lines?*
The practical use of those conclusions has been explained thoroughly at the end.
☐ **Page 22. Line 558-559**. *Could you explain how the gradient of Tm is evaluated according to Equation (16)?*

This has been done, adding this text:

'..the gradient of $T_m$ can be evaluated according to Equation (19) using temperature instead of concentration as variable.'

☐**Page 22. Lines 566 and 567**. *Which are the implications of your conclusion that there is a solid thermal resistance which depends on the rock – fracture size ratio?*
The implications are that subsurface reservoir formations with large, poorly connected, porous matrix blocks will be the optimal geological formations to be exploited for geothermal power development because isolated permeable joints will tend to lead to the distribution of heat throughout the matrix.

**Section "Conclusions"**
☐ **Page 23. Lines 602 and 603.** *Could you describe in more detail the optimal conditions for thermal exchange in a fracture network and your future research***?**
The following text has been added:

'The Explicit Network Model is an efficient computation methodology to represent flow, mass and heat transport in fractured media, as 2D and/or 3D problems are reduced to resolve a network of 1D pipe elements. Unfortunately in field case studies it is difficult to obtain the full knowledge of the geometry and parameters such as the orientations and aperture distributions of the fractures needed by the ENM even by means of field investigation methods. However in real case studies the ENM can be coupled with continuum models in order to represent greater discontinuities respect to the scale of study that generally give rise to preferential pathways for flow, mass and heat transport.

This study has permitted to detect the key parameters to design devices for heat recovery and heat dissipation that exploit the convective heat transport in fractured media.

Heat storage and transfer in fractured geological systems is affected by the spatial layout of the discontinuities.

Specifically, the rock – fracture size ratio which determines the matrix block size is a crucial element in determining matrix diffusion on fracture – matrix surface.

The estimation of the average effective thermal conductivity coefficient shows that it is not efficient to store thermal energy in rocks with high fracture density because the fractures are surrounded by a matrix with more limited capacity for diffusion giving rise to an increase in solid thermal resistance. In fact, if the fractures in the reservoir have a high density and are well connected, such that the matrix blocks are small, the optimal conditions for thermal exchange are not reached as the matrix blocks have a limited capability to store heat.

On the other hand, isolated permeable fractures will tend to lead to the more distribution of heat throughout the matrix.

Therefore, subsurface reservoir formations with large porous matrix blocks will be the optimal geological formations to be exploited for geothermal power development.

The study could help to improve the efficiency and optimization of industrial and environmental systems, and may provide a better understanding of geological processes involving transient heat transfer in the subsurface.

Future developments of the current study will be carrying out investigations and experiments aimed at further deepening the quantitative understanding of how fracture arrangement and matrix interactions affect the efficiency of storing and dissipation thermal energy in aquifers. This could be achieved by means of using different formations with different fracture density and matrix porosity.'

**Technical comments**

*There is not section numbering. Number all sections and subsections as follows:*
*1 Introduction*
*2 ……………*
*2.1 …………*
*2.2 ……*

*3 ……………*
All sections and subsections have been numbered.
☐*Be careful with the use of subscripts. For a given magnitude, sometimes you use subscripts and other not (for example, Da). Revise the whole text.*
The whole text has been revised.

☐*You would write equations from (39) to (53) in a larger size.*
The equations from (39) to (53) have been rewritten in a larger size.

**Abstract**
☐ **Page 1, Lines 13.** *Add or: "cooling of industrial processes, food drying systems **or** desalination".*
'or' has been added.

**Introduction**
☐**Page 6.** Add a new paragraph at the end of the Introduction section to summarize your paper as follows: *"In section 1 we shows.....Section 2 describe....".*
The end of the introduction has been modified so as to mention each section of the paper:
'Section 1 shows a short review about mass and heat transport in fractured media highlighting what is still unresolved or contrasting in the literature.
In Section 2 the theoretical background related to non linear flow, solute and heat transport behavior in fractured media has been reported.
…. Section 3 shows the thermal tracer tests carried out on an artificially created fractured rock sample that has been used in previous studies to analyze nonlinear flow and non Fickian transport dynamics in fractured formations (Cherubini et al., 2012, 2013a, 2013b, 2013c and 2014).
In Section 4 have been reported the interpretation of flow and transport experiments together with the fitting of BTCs and interpretation of estimated model parameters. In particular, the obtained thermal BTCs show a more enhanced early arrival and long tailing than solute BTCs.
Section 5 reports some practical applications of the knowledges acquired from this study on the convective heat transport in fractured media for exploiting heat recovery and heat dissipation.'

**Section "Theoretical background".**
**Subsection "Nonlinear flow"**
☐**Page 7, Line 186.** *Add the meaning of the coefficients $\mu$, $u_f$, $k$, $\wp$ and $p$.*

The following text has been added with the requested explanation:
'Where $x$ (m) is the coordinate parallel to the axis of the single fracture (*SF*), $p$ ($ML^{-1}T^{-2}$) is the flow pressure, $\mu$ ($ML^{-1}T^{-1}$) is the dynamic viscosity, $k$ ($L^2$) is the permeability, $u_f$ ($LT^{-1}$) is the convective velocity, $\rho$ ($ML^{-3}$) is the density.. '

☐**Page 7, Line 189.** *You write: "The coefficients a (TL-1) and b (TL-2) represent…." I think that is not a and b but a' y b'.*
a' and b' have been substituted as requested.

**Section "Theoretical background".**
**Subsection "Heat transfer by water flow in single fractures"**
☐**Page 8, Line 217.** *In equation (8) you write $C_m$ in capital letter but in equation (6) you write $c_m$ in lower case letter. Are they the same coefficients, $C_m$ and $c_m$, or are different? If they are the same, use the same notation (in capital letter or in lower case letter always). The same applies to the equations (10) and (12). The same applies to lines 247 and 251 of page 9 or in equation (14).*
This has been done; $c_m$ has been used.

☐**Page 9, Line 231**. *If you define $u_f$ before, in line 186 of page 7, you must eliminate it in line 231 of page 9.*
This has been done.

☐**Page 9, Line 244**. *You write: " (-) the matrix porosity". What is the symbol (-)?*
The symbol (-) means dimensionsless and has been eliminated.

☐**Page 9, Line 248.** *You write: "function of time in Laplace space.". Eliminate the point and write something like "as follows".*
'As follows:' has been added.

☐**Page 9, Line 250**. *After this line, write another one to define the magnitudes s, $\nu$, L, $\beta$, A and B.*
The following text has been added to define s and the equations have been put before :
Where *s* is the integral variable of the Laplace transform, *L* (L) is the length of *SF*, the *v, A, β²* and *B* coefficients are expressed as follows:

$$v = \frac{u_f}{2D_f} \tag{1}$$

$$A = \frac{\delta}{\sqrt{\theta_m D_e}} \tag{2}$$

$$\beta^2 = \frac{4D_f}{u_f^2} \tag{3}$$

$$B = \frac{1}{\sqrt{D_e}} \tag{}$$

☐**Page 9, Line 251 and Page 10, Line 252.** *Write these two lines later and after equation (19).*
This has been done, as written in the former comment.

☐**Page 10, Lines 253 to 257.** *Write these lines before and after equation (14) in line 250.*

This has been done.

$$A = \frac{\delta}{\sqrt{\theta D_e}}; \quad \theta = \theta_m$$

*Why not write*

$$A = \frac{\delta}{\sqrt{\theta D_e}}; \quad \theta = \theta_m$$

The equation has been modified into $A = \dfrac{\delta}{\sqrt{\theta_m D_e}}$

**Page 10, Lines 258.** *You write: "Furthermore on the basis of these analytical solutions the probability density function of the solute residence time (PDF)". Change this line as follows: "Furthermore on the basis of these analytical solutions the probability density function (PDF) of the solute residence time."*

This has been changed as requested.

This has been done $C_w$ has been used (capital letter)

**☐Page 11, Lines 278.** *Eliminate the semicolon at the end of equation (25).*

The semicolon has been eliminated.

$$A = \frac{\delta}{\sqrt{\theta D_e}}; \quad \theta = \frac{\rho_m C_m}{\rho_f C_f}, \quad D_e = \frac{k_e}{\rho_w C_w}$$

For example: A=…… where θ=…. and De=….

This has been done:

$$A = \frac{\delta}{\sqrt{\theta D_e}} \tag{4}$$

where $\theta = \rho_m C_m / \rho_w C_w$ and $D_e = k_e / \rho_w C_w$.

**☐Page 12, Lines 294 to 298.** *First, define Damköhler number and use equation (32). Second, write your line "Note that the inverse of te……" and define α with equation (31). Again, be careful with subscript: you write Da in equation (32) and Da in line 296, 299, 302, 303…..In equation (33), no subscripts, etc.*

This has been done as requested:

'Another useful dimensionless number, generally applied in chemical engineering, is the Damköhler number that can be used in order to evaluate the influence of matrix diffusion on convection phenomena. *Da* relates the convection time scale to the exchange time scale.

$$Da = \frac{t_u}{t_e} = \frac{\alpha L}{u_f}$$ (5)

Where $\alpha$ (T$^{-1}$) is the exchange rate coefficient corresponding to:

$$\alpha = \frac{D_e}{\delta^2}$$

**Section "Theoretical background".**
**Subsection "Explicit network model"**
☐**Page 12, Line 312 and Page 13, Lines 316 and 317-318.** *Use another notation to write SF j (this is a bit confusing). Something like j-th simple fracture or j-th SF.*
This has been done:
'With the assumption that a $j^{th}$ SF can be schematized by a 1D – pipe element, the Forchheimer model can be used to write the relationship between head loss $\Delta h_j$ (L) and flow rate $Q_j$ (L$^3$T$^{-1}$) in finite terms'.

'…Where $L_j$ (L) is the length of $j^{th}$ SF, $a$ (TL$^{-3}$) and $b$ (T$^2$L$^{-6}$) represent the Forchheimer parameters written in finite terms. The term in the square brackets constitutes the resistance to flow $R_j(Q_j)$ (TL$^{-2}$) of $j^{th}$ SF.'

☐**Page 12, Line 312.** *You write "schematized by a 1d – pipe element". Use better 1 D-pipe element.*
1D has been substituted

☐**Page 13, Lines 319 and 322.** *Write 2D instead of 2d.*
2D has been substituted

☐**Page 14, Line 339.** *Write the convolution operator without parenthesis: *.*
This has been done.

☐**Page 14, Line 340.** *The notation for the inverse Laplace transform operator, L-1, could be mistaken for the characteristic length, L. Use another notation.*
$\mathcal{L}^1$ has been used.

☐**Page 14, Lines 348-349.** *Modify the phrase as follows: Where T0 (K) is the initial temperature, Tinj (K) is the temperature injection function and PH,j is the heat distribution probability.*
The sentence has been modified as requested.

**Section "Material and methods"**
**Subsection "Description of the experimental apparatus"**
☐**Page 15, Lines 375.** *Verify the writing of the units: µS cm-1.*
(µS cm$^{-1}$) has been corrected.

**Section "Results and discussion"**
**Subsection "Flow characteristics"**
☐**Page 16, Line 406.** *Write 2D instead of 2d.*
2D has been substituted.

☐**Page 17. Line 417.** *Write Fo=1 in italics.*
*Fo* has been written in italics.

**Section "Results and discussion"**
**Subsection "Fitting of breakthrough curves and interpretation of estimated model parameters"**
☐**Page 17. Line 430.** *It is better to indicate the number of the section you refer, and not "the previous section".*
'presented in section 2.3' has been added.

☐**Page 17. Line 433**. *Review the number of the equations quoted: Are not equations 36 and 37?*
This has been modified as requested.

☐**Page 17**. **Lines 441 to 444**. *You would quote Tables 1 and 2 in this paragraph to verify the conclusions deduced in these lines.*
Tables 1 and 2 have been quoted for the whole paragraph 'The results presented in Tables 1 and 2 highlight that:...'

☐ **Page 18**. **Lines 448 to 451**. *The same as before: you would quote Tables 1 and 2 in this paragraph to verify the conclusions deduced in these lines.*
Tables 1 and 2 have been quoted for the whole paragraph 'The results presented in Tables 1 and 2 highlight that:...'

☐**Page 18. Line 468**. *You write: "whereas uf is influenced". Add a blank space.*
A blank space has been added as requested.

**References**
*Review all the bibliographic references.*
*Review the NPG format for references because there are several references that are not well written (the punctuation, the initial page or the final page, the position of the year is not at the end or the review is not written in italics). For example, the following references are not correct:*
☐*Auradou, H., Deazerm G., Boschan, A., Hulin J., Koplik, J., 2006. Flow channeling in a single fracture induced by shear displacement. Geothermics 35 575-588.*
☐*Becker, M.W. and Shapiro, A. M., 2003. Interpreting tracer breakthrough tailing from different forcedgradient tracer experiment configurations in fractured bedrock. Water Resources Research. 39(1):1024. pages.*
☐*Cherubini, C., Pastore, N., 2011. Critical stress scenarios for a coastal aquifer in southeastern Italy. Natural Hazards and Earth System Science. 11 (5) p. 1381-1393.*

All references have been rewritten according to NPG format.

*And more.*
*I do not understand the following reference:*
*Geiger, S. and Emmanuel, S., 2010. Non-fourier thermal transport in fractured geological media. Water Resources Research, 46, xvii, 26, 27, 168.*
The reference has been modified:

Geiger, S. and Emmanuel, S.: Non-fourier thermal transport in fractured geological media. Water Resources Research, Vol 46,W07504, doi:10.1029/2009WR008671, 2010.

*The following references are not quoted in the text but they are in the list:*
☐*De Hoog, F.R., Knight, J.H., Stokes, A.N., 1982. An improved method for numerical inversion of Laplace transforms. SIAM J. Sci. Stat. Comput. 3 (3), 357-366.*
☐*Hasler, A., Gruber, S., Font, M., Dubois, A., 2011. Advective heat transport in frozen rock cleft – Conceptual model, laboratory experiments and numerical simulation…..pages?*

Those references have been deleted in the list because not quoted in the text.

*You quote in the text some references but there are no corresponding bibliographic references in the list:*
☐*Neuville et al. (2010)*

*All quotes in page 4 are not in the reference list.*
☐*Hopmans et al. (2002).*
☐*Sauty et al. (1982)*
☐*Papadopulos and Larson, 1978.*
☐*Smith and Chapman, 1983.*
☐*Molson et al., 1992.*
☐*De Marsily, 1986.*
☐*Anderson, 2005.*
☐*Hatch et al., 2006.*
☐*Keery et al., 2007.*
☐*Vandenbohede et al., 2009.*
☐*Vandenbohede and Lebbe, 2010.*
☐*Rau et al., 2010.*
☐*Green et al., 1964.*
☐*Wu et al., 2010.*

All those references have been inserted.

*You mention the following reference (Cherubini et al., 2013) in the text (page 9, line 233 and page 18, line 457) but you write four different references in the list: 2013a, 2013b, 2013c and 2013d. Which is the reference you want to quote? All?*

The correct references have been inserted in the text:
Cherubini et al., 2012, 2013a, 2013b, 2013c and 2014 for page 9, line 233
Cherubini at al., 2013a, 2013b, 2013c and 2014 for page 18, line 457

*Use the same line spacing in the reference list (see page 25 from line 657 to 668 and page 26 from line 669 to 674).*
The same line spacing has been used

*The following reference (page 26, line 673 to 674):*
*Martinez, A. R., Roubinet, D., Tartakovsky, D. M., 2014. Analytical models of heat conduction in fractured rocks. J. Geophys. Res. Solid Earth, 119.*
*is not in a separate line from the previous bibliographic reference.*
The reference has been separated

**Tables**
☐Table 1 and 2
*Modify the caption to include the definition of $Q_o$, $u_f$, $D_f$ and $\alpha$.*
The caption has been modified with the definitions *of $Q_o$, $u_f$, $D_f$ and $\alpha$.*

***Figures***
☐**Figure 1.** *It would be interesting to add a photograph of the experimental setup in Figure 1.*
A photo of the experimental setup has been added:

Figure 1. a) fractured block sealed with epoxy resin. b) thermal insulated fracture block connected to the hydraulic circuit.

☐**Figure 2.** *Modify the caption to include the definition of $Q_o$, $Q_1$ and $Q_2$. Is it possible to use a frontal figure and not this tilted one? May be, it is also possible to say that is an extension of the rock matrix in Figure 1.*

The caption has been modified to include the mentioned definitions: 'Two dimensional pipe network conceptualization of the fracture network of the fractured rock block in Figure 1. $Q_0$ is the injection flow rate, $Q_1$ and $Q_2$ are the flow rates that flowing in the parallel branch 6 and 3-4-5 respectively.'

The figure has been modified into frontal and the sentence 'conceptualization of the fracture network of the fractured rock block in Figure 1' says that it is an extension of the rock matrix in Figure 1

☐**Figure 3.** *The label of the Y-axis in the upper left figure is different from the others Y-labels.*
The label of Y-axis in the upper left figure has been modified to be the same as the other Y-labels.

☐**Figure 4**. *Describe the meaning of the blue curve in the caption.*
The meaning of all the curves has been explained and the caption has been modified into: 'Fitting of BTCs at different injection flow rates using ENM with Tang's solution for heat transport. The blue curve is the temperature observed at the inlet port used as the temperature injection function, the red square curve is the observed temperature at the outlet port, the black continuous curve is the simulated temperature at the outlet port.'

☐**Figure 7** *is not quoted in the text.*
Figure 7 has been quoted in the text: 'Figure 7 shows the exchange rate coefficient $\alpha$ as function of the convective velocity $u_f$ for both mass and heat transport.'

*Figure 12. Review the writing of the symbols $D_a$ and $Q_o$ with and without subscripts in the text, in the caption of this figure and in the labels of the axis.*

The writing of Da and $Q_0$ has been reviewed. Da and $Q_0$ have been used everywhere in the paper.

*Figure 15. Use smaller symbols as in the previous figures.*

Smaller symbols have been used as in the previous figures.

---

## Author Comment (AC2) · 6 Dec 2016

*Overall, the paper tells a good story. The modeling and experiments is well described and conclusions drawn are reasonable.*

*In the conclusions, the authors should say a few words about future directions. What about more complex fracture networks? What are the range of pore sizes and porosity and permeability that the ENM can be applied to?*

The conclusion has been widely extended and future directions have been better addressed.

This text has been added:

'The Explicit Network Model is an efficient computation methodology to represent flow, mass and heat transport in fractured media, as 2D and/or 3D problems are reduced to resolve a network of 1D pipe elements. Unfortunately in field case studies it is difficult to obtain the full knowledge of the geometry and parameters such as the orientations and aperture distributions of the fractures needed by the ENM even by means of field investigation methods. However in real case studies the ENM can be coupled with continuum models in order to represent greater discontinuities respect to the scale of study that generally give rise to preferential pathways for flow, mass and heat transport.

This study has permitted to detect the key parameters to design devices for heat recovery and heat dissipation that exploit the convective heat transport in fractured media.

Heat storage and transfer in fractured geological systems is affected by the spatial layout of the discontinuities.

Specifically, the rock – fracture size ratio which determines the matrix block size is a crucial element in determining matrix diffusion on fracture – matrix surface.

The estimation of the average effective thermal conductivity coefficient shows that it is not efficient to store thermal energy in rocks with high fracture density because the fractures are surrounded by a matrix with more limited capacity for diffusion giving rise to an increase in solid thermal resistance. In fact, if the fractures in the reservoir have a high density and are well connected, such that the matrix blocks are small, the optimal conditions for thermal exchange are not reached as the matrix blocks have a limited capability to store heat.

On the other hand, isolated permeable fractures will tend to lead to the more distribution of heat throughout the matrix.

Therefore, subsurface reservoir formations with large porous matrix blocks will be the optimal geological formations to be exploited for geothermal power development.

The study could help to improve the efficiency and optimization of industrial and environmental systems, and may provide a better understanding of geological processes involving transient heat transfer in the subsurface.'

Future developments of the current study will be carrying out investigations and experiments aimed at further deepening the quantitative understanding of how fracture arrangement and matrix interactions affect the efficiency of storing and dissipation thermal energy in aquifers. This could be achieved by means of using different formations with different fracture density and matrix porosity.

*Minor point: the quality of the figures should be improved.*

The quality of all figures has been improved in terms of better resolution. Two figures (photos) have been added to show the experimental setup.